# Single-molecule imaging reveals distinct elongation and frameshifting dynamics between frames of expanded RNA repeats in C9ORF72-ALS/FTD

Malgorzata J. Latallo[1,2,9], Shaopeng Wang[1,2,3,4,9], Daoyuan Dong[3,4,9], Blake Nelson [1,2], Nathan M. Livingston [1,2], Rong Wu[3,4], Ning Zhao[5], Timothy J. Stasevich [5], Michael C. Bassik [6], Shuying Sun [2,3,4,7,8] ✉ & Bin Wu [1,2,7] ✉

*C9ORF72* hexanucleotide repeat expansion is the most common genetic cause of both amyotrophic lateral sclerosis (ALS) and frontotemporal dementia (FTD). One pathogenic mechanism is the accumulation of toxic dipeptide repeat (DPR) proteins like poly-GA, GP and GR, produced by the noncanonical translation of the expanded RNA repeats. However, how different DPRs are synthesized remains elusive. Here, we use single-molecule imaging techniques to directly measure the translation dynamics of different DPRs. Besides initiation, translation elongation rates vary drastically between different frames, with GP slower than GA and GR the slowest. We directly visualize frameshift events using a two-color single-molecule translation assay. The repeat expansion enhances frameshifting, but the overall frequency is low. There is a higher chance of GR-to-GA shift than in the reversed direction. Finally, the ribosome-associated protein quality control (RQC) factors ZNF598 and Pelota modulate the translation dynamics, and the repeat RNA sequence is important for invoking the RQC pathway. This study reveals that multiple translation steps modulate the final DPR production. Understanding repeat RNA translation is critically important to decipher the DPR-mediated pathogenesis and identify potential therapeutic targets in C9ORF72-ALS/FTD.

The most prevalent genetic cause of amyotrophic lateral sclerosis (ALS) and frontotemporal dementia (FTD) is the hexanucleotide repeat expansion of GGGGCC in the non-coding region of the first intron of the *C9ORF72* gene[1–3]. The repeat is typically fewer than ten in healthy individuals but can expand up to thousands in C9ORF72-ALS/FTD patients[4,5]. The C9ORF72 repeat expansion may cause RNA-mediated gain-of-toxicity: formation of RNA granules disrupting the RNA processing, as well as production of toxic dipeptide repeat (DPR) proteins[6–9]. The DPRs are synthesized through noncanonical repeat-associated non-AUG (RAN) translation[10,11]. Due to bidirectional transcription in the *C9ORF72* locus, both sense and antisense RNA repeats exist, which are translated to five different DPRs: GA, GP, GR, PR and PA[10,12,13]. Numerous studies have revealed the biophysical properties and toxicity pathways of different DPRs[14,15]. The arginine-containing DPRs (poly-GR and poly-PR) are the most toxic[14,16–27], however, poly-GA and poly-GP aggregates are more widely observed in patient post-mortem tissues[13,28]. Chimeric DPR species consisting of fused GA and GP/GR were detected in patients, but it is hard to quantify the relative

abundance. They can be produced either by frameshift during translation or due to genetic interruptions in the expanded repeats[29–31]. To better understand the relative contribution of the different DPRs to the disease, it is imperative to thoroughly study the DPR biosynthesis and translation dynamics of different reading frames[14].

Protein synthesis is an elaborate process, subject to tight regulation at each step. Translation is divided into four stages: initiation, elongation, termination, and ribosome recycling[32]. Any one of the steps may contribute to the final protein production. For RAN translation, most studies focused on the initiation, commonly believed to be the rate-limiting step in determining translation efficiency. Several translation initiation factors have been shown to regulate GGGGCC-associated RAN translation, including eIF2A[33], eIF2D[34], eIF4B and eIF4H[34,35]. The phosphorylation of eIF2α, the key factor in the integrated stress response (ISR) pathway that generally reduces the global translation, can significantly enhance RAN translation[36,37]. For the cis-acting elements that regulate RAN translation, it was shown that the sequence preceding the repeats plays an important role in initiation. Particularly, the CUG codon, 24 nucleotides (nt) upstream of the repeats, is a near-cognate start codon used for the initiation in GA frame[37,38]. Therefore, the translation efficiency from the GA frame is higher than GP and GR frames, which could account for the different DPR abundance in patients. However, it remains elusive whether initiation alone can explain different DPR expressions and how translation elongation and frameshift contribute.

Translation efficiency is often measured experimentally as the ribosome occupancy density[39,40], which is proportional to the initiation rate and inversely related to the elongation rate[41]. Recently, it has been increasingly recognized that translation elongation also plays a regulatory role[42]. On the one hand, optimal codon usage, which presumably promotes faster codon adaptation and elongation, is known to increase protein yield[43–45] and associate with mRNA stability[46]. On the other hand, stalling sequences, such as strong RNA secondary structures or stretches of polybasic amino acids, inhibit ribosome elongation, induce ribosome collisions and invoke protein/RNA quality control pathways to degrade nascent peptides and/or mRNAs[47–49]. This is known as the ribosome-associated protein quality control (RQC)[50]. The GGGGCC repeat is prone to form G-quadruplexes or strong hairpin structures[51,52], which could potentially inhibit ribosome elongation. Poly-GR is highly positively charged, and poly-GP is proline-rich and may induce steric hindrance. These DPRs are strong candidates for ribosome stalling[53] and targets for RQC[54,55]. How they influence translation elongation dynamics is largely unknown.

Traditionally, DPR biosynthesis is studied through ensemble methods to measure the final protein level, which depends on the RNA transcription, processing, export, translation, degradation, and protein stability. The final protein amount may not precisely represent the translation efficiency. Here, we employed the SunTag-based single-molecule translation imaging approach developed by us and other laboratories[56–60] to address the mechanistic questions of DPR translation that are difficult to address with ensemble methods. The single-molecule imaging technique directly measures the temporal translation dynamics of single mRNAs in live cells. It can reveal the molecular heterogeneity hidden in the ensemble measurements. We constructed reporters to measure initiation, elongation, and frameshifts for different DPR frames. We mutated the CUG near-cognate start codon to examine its influence on the initiation efficiency in each frame. We generated a two-color translation reporter to observe frameshifting events on single RNAs. We measured the elongation speed in different frames and characterized its dependence on the amino acid sequence and codon usage. Lastly, we characterized how RQC factors were involved in modulating the translation dynamics through repeats. The insights gained from this single-molecule study are instrumental in understanding the biogenesis of DPR and their

contribution to the disease. Modulation of translation is a promising therapeutic approach to prevent the accumulation of toxic DPRs.

## Results

### The CUG near-cognate start codon contributes to the initiation of the GA frame, but not GP or GR frame

To compare RAN translation initiation efficiency among different reading frames, we constructed single-molecule imaging reporters for each frame (Fig. 1a): 70× (GGGGCC) repeats, including the 113-nucleotide upstream intron sequences, were placed in front of 24× SunTag without the AUG start codon. Zero, one or two nucleotides were inserted between the (GGGGCC) expansion and SunTag to make SunTag in frame with a particular frame (GA, GP or GR). SunTag epitope is followed by an auxin-inducible degron (AID), which degrades mature proteins upon the addition of auxin to reduce the background fluorescence[61,62]. 24× MS2-binding sites (MBS) were inserted in the 3' untranslated region (UTR) to visualize the RNA[63]. Next, each of these constructs was stably integrated into the U-2 Osteosarcoma (U-2 OS) cell line expressing stdMCP-HaloTag (synonymized tandem MS2 coat protein fused with HaloTag) for RNA labeling[63,64], and scFv-sfGFP (single-chain antibody fused with superfolder GFP) for protein visualization[65]. We performed single-molecule Fluorescence In Situ Hybridization (smFISH), with probes targeting the MBS and SunTag regions, and immunofluorescence (IF) with antibodies against GFP[58] in these cells to quantify the nascent peptides on mRNAs. Without RNA reporters, scFv-sfGFP itself does not form detectable green puncta (Supplementary Fig. 1a, b). With RNA reporters, both red (RNA) and green puncta (protein) were observed in the cytoplasm. Dim green puncta in the cytoplasm represented the fully synthesized proteins since they could bind up to 24 scFv-sfGFPs and were readily detectable. Bright green puncta colocalized with RNA were translation sites (TLS, arrow) as one or more ribosomes were translating on the mRNA to produce multiple nascent peptides (Fig. 1b). To compare the RAN translation initiation efficiency in different frames, we quantified the percentage of translating mRNA in each reporter cell line. The repeats increased the fraction of translating mRNA (RAN translation, 10–20%) significantly compared with negative control (without insertion of repeats, <2%) (Fig. 1c, Supplementary Fig. 1c). The RAN translation was significantly less efficient than regular mRNAs (~80%)[58] (Fig. 1c), as expected. Interestingly, the fraction of translating mRNA in the GA frame was higher than that of GP and GR frames (Fig. 1c), supporting the higher initiation rate of poly-GA.

It has been reported that a near-cognate start codon CUG 24 nt upstream of GGGGCC repeat expansions is the initiation site of poly-GA[37,38]. To measure the influence of CUG on translation initiation, we made reporters in which the CUG was mutated to CGG (Fig. 1a). We performed live-cell imaging to quantify the translation dynamics of single mRNAs. To facilitate long-term tracking, mRNAs were tethered to the cell plasma membrane to limit their mobilities (by fusing stdMCP to a CAAX motif)[56] and imaged with total internal reflection fluorescence (TIRF) microscopy to reduce background noise. Cells were imaged to track translation of single mRNAs for 30 min (Fig. 1d, e, Supplementary Movies 1 and 2). The fraction of time that an individual mRNA molecule was being translated was quantified. In the GA frame, the RNA was translated around 45% of the time, similar to the GP frame (~40%) but significantly higher than the GR frame (~17%) (Fig. 1f). Mutation of CUG to CGG caused a 2-fold reduction in the fraction of translation time in GA frame; but did not significantly change in the GP and GR frame (Fig. 1f). The influence of CUG on different reading frames can also be validated using bicistronic luciferase reporters with the splicing context of C9ORF72[66]. The NanoLuc luciferase was placed in the first intron after the repeats in different reading frames to measure the corresponding DPR levels, while Firefly luciferase was put in exon 2 to normalize the expression level (Supplementary Fig. 1d). GA

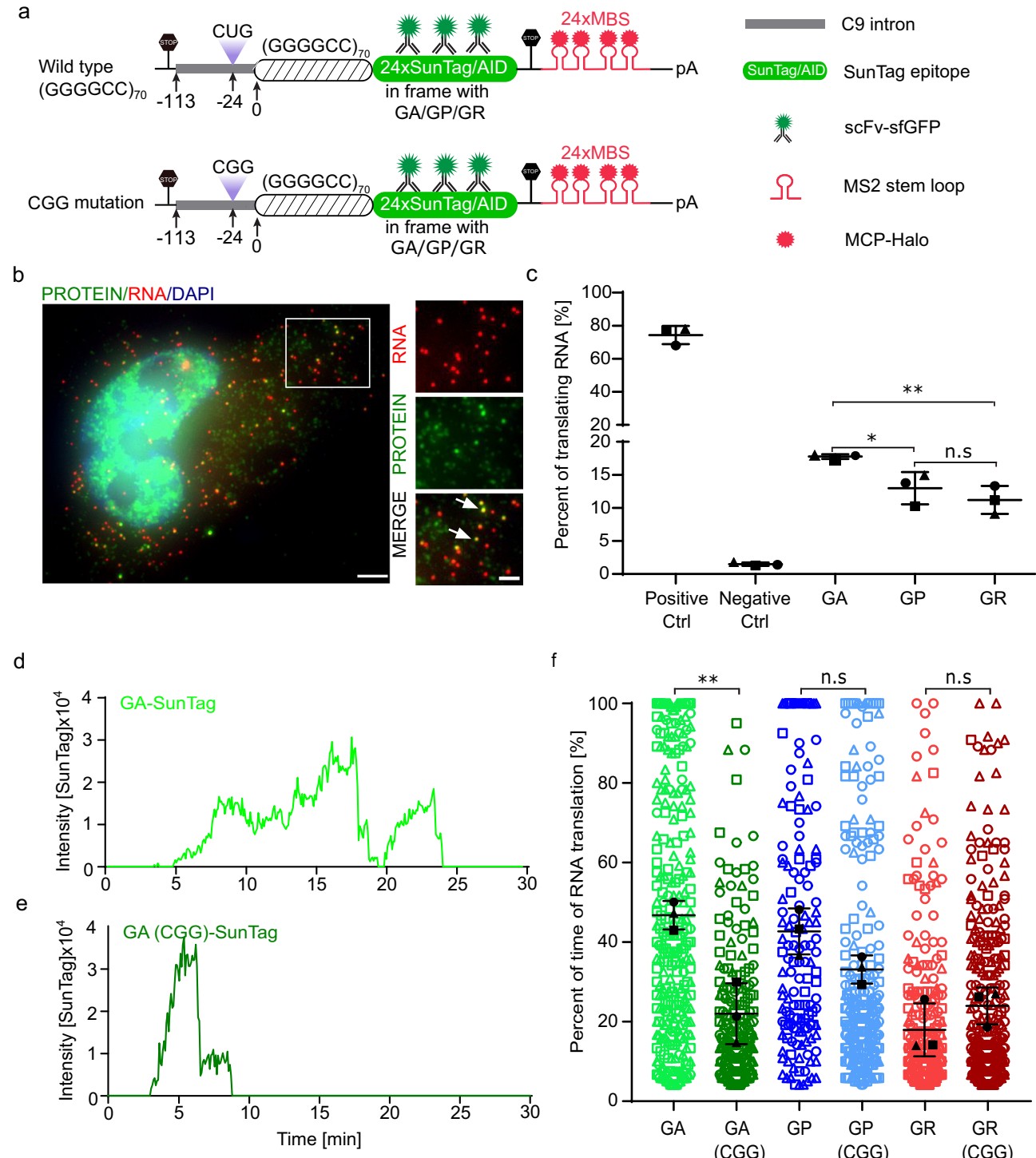

frame has significantly higher expression than GP and GR. When CUG was mutated to CGG in the intron, the GA-NLuc was significantly reduced, while there was no decrease in GP and GR (Supplementary Fig. 1e).

Taken together, both the fixed, live cell and bulk experiments suggest that RAN translation initiation efficiency in poly-GA is higher than poly-GP and poly-GR. This is due to the presence of the near-cognate CUG start codon in the GA frame. The poly-GP and poly-GR translation likely initiate independently from the CUG codon (otherwise, we should expect poly-GP and poly-GR to decrease in CGG mutant as well). Surprisingly, the difference between frames, especially between GA and GR, measured by the

luciferase assay is higher than that in the single-molecule measurements (Fig. 1c, Supplementary Fig. 1e), prompting us to look at other aspects of translation that may contribute to the variation in DPR levels.

## Frameshift occurs, but infrequently during the translation of C9ORF72 repeats

It was suggested that the chimeric DPR species existed in C9ORF72-ALS/FTD patients, either by translation frameshift or genetic interruptions of repeats[29]. Although the translation initiation reporter data suggest that the majority of poly-GP and poly-GR translation is not due to frameshifting from the GA frame, it is intriguing to directly assess

**Fig. 1 | Single-molecule imaging of RAN translation showed higher initiation rate of poly-GA than poly-GP and poly-GR. a** Schematic of constructs for single-molecule imaging of RAN translation. The wild-type constructs contain 113 bp endogenous 5′ intron sequence upstream of the GGGGCC repeats. A CUG near-cognate start codon (−24 bp) is in frame with GA. Multiple stop codons were placed at the beginning of the reporters to prevent any leakage from canonical translation. SunTag epitopes were fused with GGGGCC repeats in each of the three frames separately. In CGG mutants, the CUG codon was mutated to CGG. MBS MS2-binding sites, MCP MS2 coat proteins, scFv-sfGFP single-chain variable fragment against SunTag epitope fused with superfolder GFP, AID auxin-inducible degron. **b** Representative smFISH-IF image of RAN translation reporters. The protein signals colocalized with RNA indicated the active translating RNA. Red: RNAs; Green: protein. Scale bar: 5 μm. **c** The percentage of actively translating RNAs measured from smFISH-IF experiments. Data are mean ± SD from three biological replicates

(Ctrl:86, 39, 40 cells; Negative Ctrl: 66,60,58 cells; GA: 82, 67, 29 cells; GP: 51, 33, 26 cells; GR: 61, 53, 25 cells). Each symbol represents the average of one biological replicate. Statistical analysis using two-tailed, equal variance, t-test: GA vs GP, *P = 0.028; GA vs GR, **P = 0.0061; GP vs GR, P = 0.39. **d, e** Representative traces of translation site intensity for a single mRNA in GA (**d**) and GA-CGG mutant (**e**) reporters. Also, see Supplementary Movies 1 and 2. **f** The fraction of time that RNAs were undergoing translation during the 30 min. Each symbol represented a single translating RNA, and the shapes represented biological replicates (numbers of translating RNA traces in replicates: GA: 43, 150, 100; GA-CGG: 81, 91, 49; GP: 60, 30, 52; GP-CGG: 77, 35, 116; GR: 103, 98, 99; GR-CGG: 170, 33, 119). The mean of each replicate (black shapes) was used to calculate the mean (horizontal bar) and standard deviation (error bars) in each group. Statistical analysis using two-tailed, equal variance t-test: GA vs GA(CGG), **P = 0.0071; GP vs GP(CGG), P = 0.070; GR vs GR(CGG), P = 0.27. Data are mean ± SD.

whether the frameshifting can occur on the repeats during translation and, if it does, how frequently it occurs.

Ribosomal frameshifting is an evolutionarily conserved biological process found in many organisms[67]. Frameshift can be accidental as well as programmed. For example, ribosome stalling due to secondary RNA structure may induce a ribosome to shift its reading frame[68]. To directly visualize the frameshifting event, we utilized a dual-color translation imaging assay to measure the translation in two reading frames simultaneously on single RNAs in live cells[56,69]. We made constructs in which 70× (GGGGCC) repeats were placed between 24× synonymized HA-tag (Human influenza hemagglutinin) and 24× SunTag. The HA-tag was initiated with AUG start codon and placed before the GGGGCC repeats in one reading frame (DPR1), followed by the stop codon. The SunTag lacking AUG start codon was fused after the repeats in +1 or +2 reading frame (DPR2) with respect to DPR1 and followed by an in-frame stop codon (Fig. 2a). The mRNAs were labeled with 24× MBS in 3′UTR and tethered to cell membrane for long-term imaging by 12× PP7 binding sites (PBS) with membrane-localized PP7 coat protein[70]. This construct allows us to quantify the frameshift events in live cells. Normally, when a ribosome initiates at the AUG start codon, translates HA-tag and DPR1 in GGGGCC repeats, and terminates before SunTag, the mRNA will contain only the HA-tag TLS signal. However, if the ribosome shifts into the DPR2 frame during the translation of GGGGCC repeats, the SunTag will be translated. In such cases, both the HA-tag and SunTag translation signal will colocalize with the mRNA (Fig. 2b).

To validate that the two tags can faithfully represent the translation of two reading frames, we first constructed a negative control reporter with no repeats between HA-tag and the +1 SunTag (Supplementary Fig. 2a) and a readthrough reporter in which HA-tag and SunTag sandwiching the 70× (GGGGCC) repeats were both placed in the GA frame (Supplementary Fig. 2a, b). We stably expressed these reporters in the U-2 OS cell line containing scFv-sfGFP and Fb-HaloTag (anti-HA Frankenbody: an scFv against the HA-tag fused with HaloTag)[71]. In the negative control reporter, there is no SunTag signal colocalized with RNA and HA-tag TLS (Supplementary Movie 3). For the readthrough reporter, the vast majority (85%) of translating mRNAs (red) colocalized with both HA-tag (magenta) and SunTag (green) signals (Supplementary Fig. 2c, d, Supplementary Movie 4). These experiments demonstrate that the two epitopes reliably report translation dynamics under our experimental conditions.

To systematically compare frameshift efficiency, we made a series of reporters to quantify frameshift events from GA to GR, GA to GP, or from GR to GA (Fig. 2c). As expected, most of the mRNAs (red) colocalized with HA-tag TLS (magenta) (Fig. 2d, Supplementary Movie 5), indicating the high translation efficiency of AUG initiated translation. In all three reporters, we observed occasional and transient SunTag fluorescence signals (green) colocalized with both mRNA and HA-tag signals during the 30-min live-cell imaging, indicating the occurrence of translational frameshift (Fig. 2d, e,

Supplementary Movie 5). This supports that the expanded repeats enhance the frameshift probability compared to the negative control. We analyzed individual translating mRNAs and quantified the frequencies of frameshift by calculating the percentage of mRNAs showing both SunTag and HA-tag signals within the imaging window (Fig. 2f, Supplementary Fig. 3a–c). First, we noticed that frameshift is rare: more than 85% of translating mRNAs in all three reporters did not have any frameshift events in 30 min (Fig. 2g). Second, the percentage of mRNAs undergoing frameshift from GR to GA (10.6%) was significantly higher than from GA to GR (2.5%) and from GA to GP (5.6%) (Fig. 2g). Similarly, we quantified the amount of time spent in particular translation state. About 90% of the time, mRNAs were translated in the original ORF without frameshifting. However, when frameshift did happen, it occurred more often for GR to GA (6.4%), followed by GA to GP (4.3%) and lastly GA to GR (2.3%) (Supplementary Fig. 3d).

We observed low numbers of mRNAs that only had SunTag signals colocalized, which likely represented cap-independent RAN translation events as reported previously[66,72]. To quantify the contribution of RAN translation, we constructed a bicistronic reporter. We placed stop codons in all three reading frames between HA-tag and repeats, such that the SunTag signal arose only from the cap-independent RAN translation in the GA frame (Supplementary Fig. 4a). We observed rare RAN SunTag TLS (Supplementary Fig. 4b, c, Supplementary Movie 6), with a frequency similar to the SunTag-only events in the frameshift reporters (about 2%, Fig. 2g). This supports that the quantification of the frameshift events is not significantly influenced by the cap-independent RAN translation and represent the relative frequency between different frames.

To further quantify the relative amount of frameshift products, we made dual-luciferase constructs in which the AUG initiated Firefly Luciferase (FLuc) was placed before the 70× (GGGGCC) repeats in the GR frame followed by the in-frame stop codon (Fig. 3a). Nanoluc Luciferase (NLuc) without an AUG start codon was placed after the repeats in either GR (0), GA (+1) or GP (+2) frame to compare frameshift efficiency. Identical constructs without (GGGGCC) repeats were used as negative controls (Fig. 3a). We transiently expressed these constructs in three different cell lines and quantified the frameshift products. Without GGGGCC repeats, no NLuc activity was detected in the +1 or +2 reading frame (Fig. 3b–d). 70× (GGGGCC) repeats significantly increased frameshift products in both +1 (GR-to-GA) and +2 (GR-to-GP) frames (Fig. 3b–d). The GR-to-GA shift was more efficient (2.3–5.3%) than the GR-to-GP shift (0.5–1.6%), despite the overall low frameshift efficiency. Taken together, both single-molecule imaging experiment and bulk luciferase data indicate that frameshift could occur during the translation of GGGGCC repeat expansion and the chimeric DPRs can be produced. However, considering the overall low frameshift frequency in multiple cell lines, chimeric DPRs likely account only for a small proportion of total DPRs.

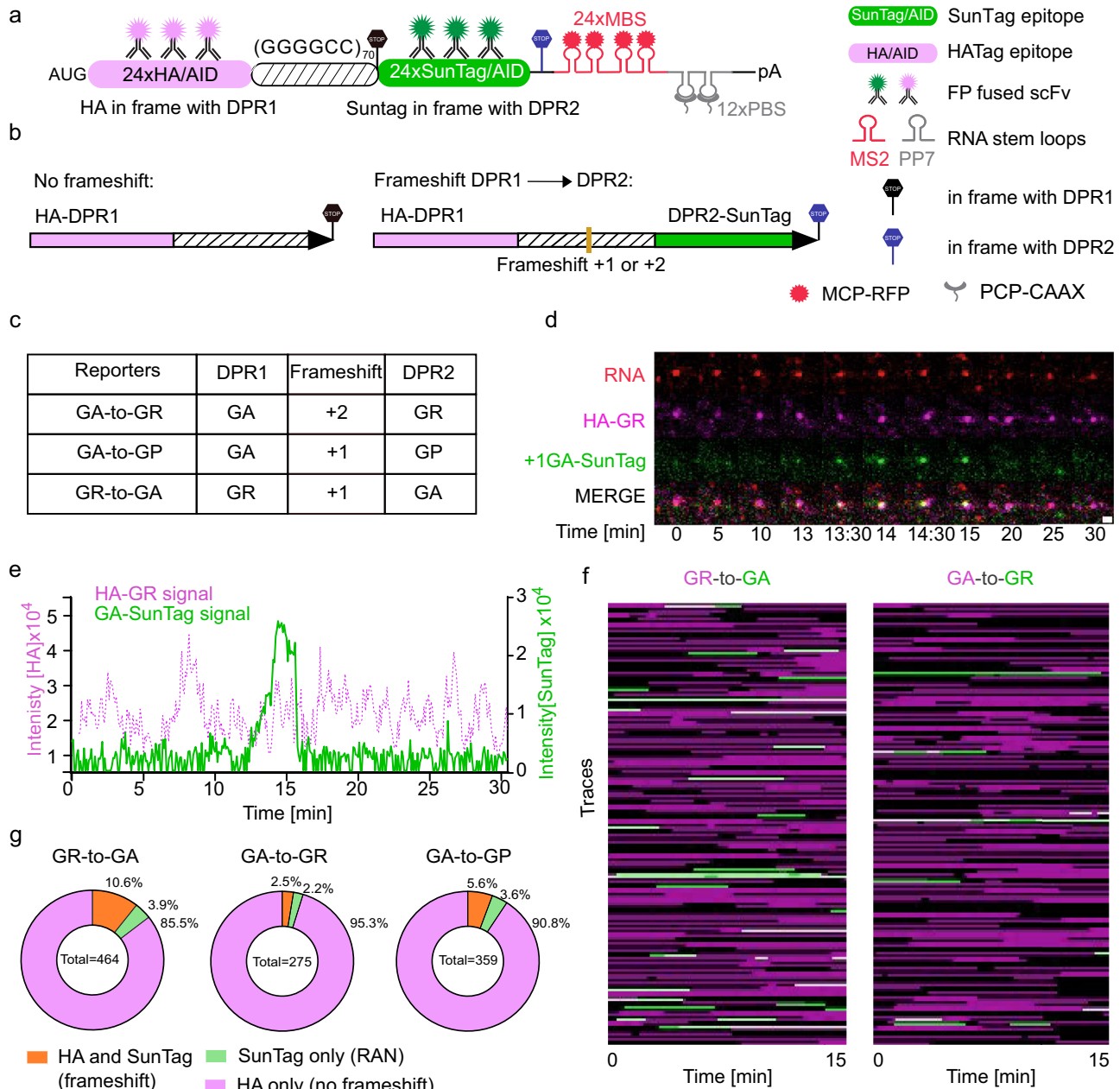

**Fig. 2 | Two-color translation assay revealed translation frameshift frequencies between DPR frames of GGGGCC repeats. a** Schematic of the dual-color frameshift constructs. 24× HA-tag were placed before 70× (GGGGCC) repeats in frame with DPR1. The translation was initiated by a canonical AUG start codon before HA and terminated with a stop codon after GGGGCC repeats. 24× SunTag lacking AUG start codon was placed after 70× (GGGGCC) repeats in the frame of DPR2. 24× MBS was used for RNA visualization and 12× PBS was used to tether mRNA on the cell membrane for long-term tracking. **b** Expected protein products without or with translation frameshift. **c** Frameshift reporters used in this study. **d** Time-lapse images illustrating translation frameshift events for GR-to-GA reporter. Red (RNA): stdMCP-RFP; magenta (GR): HA-Fb-HaloTag; green (GA): SunTag-scFv-sfGFP. See also Supplementary Movie 5. **e** Fluorescence intensity traces of both HA-tag and SunTag on a single mRNA showing frameshift events (Corresponding to (**d**)). **f** Combined TLS tracks in two translation channels for GR-to-GA and GA-to-GR frameshift reporters. Magenta: HA-Fb-HaloTag indicating the translation of AUG-HA-DPR1; green: SunTag-scFv-sfGFP indicating the translation of DPR2-SunTag. **g** Percentage of translating mRNAs undergoing normal translation (only in HA-DPR1 frame, magenta), frameshift (both frames, orange), or RAN translation (only DPR2-SunTag frame, green). The number of mRNAs under specific state was divided by the total number of tracks to calculate the percentage: GR-to-GA (5 cells, 464 translation events); GA-to-GR (6 cells, 275 of translation events); GA-to-GP (6 cells, 359 of translation events).

## The translation elongation speeds in GP and GR frames are slower than in GA frame

The GGGGCC repeat sequence involves highly repetitive codon usage and can form strong secondary structures, such as hairpin or G-quadruplex. It is not well understood how they influence the elongation of ribosomes through the repeats. Earlier, we demonstrated ribosomes translating poly-GR were more likely to frameshift. A plausible explanation for the observation is that the slower elongation speed of poly-GR leads to ribosomal collision and subsequent sliding to a different frame. We therefore examined the elongation kinetics of ribosomes through different reading frames. We constructed a series of reporters with SunTag placed before the repeats in different reading frames (Fig. 4a). After the SunTag sequence is translated, the fluorescence signal remains on the nascent peptide while the ribosome translates the repeats, therefore reporting on the elongation kinetics through the specific reading frames of the repeats. The control

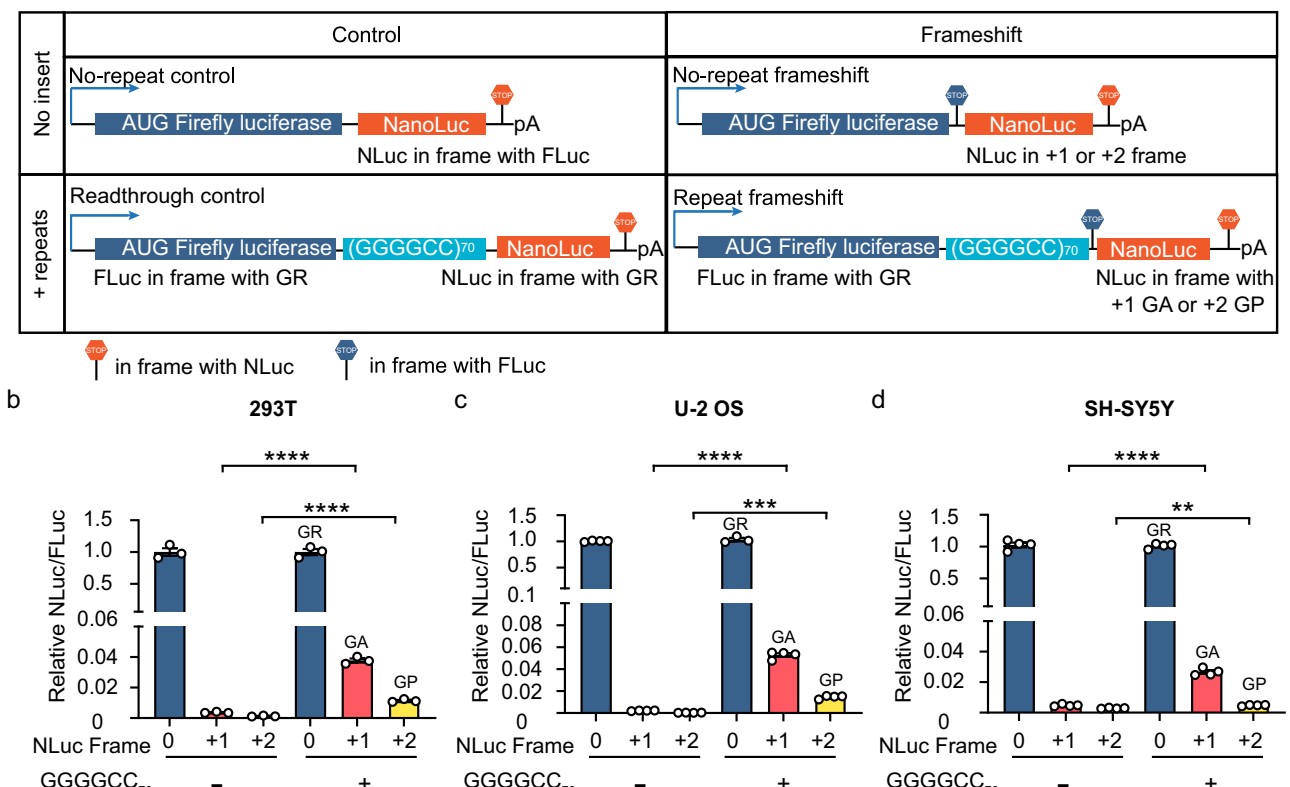

**Fig. 3 | Luciferase assay to measure proteins produced from frameshifting.**
**a** Schematic of dual-luciferase constructs used to compare frameshift products. In the no-repeat control, NLuc was fused in frame with FLuc as a single open reading frame. In the readthrough control, NLuc was placed in the same frame between FLuc and GR-(GGGGCC)$_{70}$. In the no-repeat frameshift control, NLuc was placed in the +1 or in the +2 frame, with stop codon placed in the FLuc frame before NLuc. In the repeat frameshift construct, the GR frame of (GGGGCC)$_{70}$ was placed in frame with FLuc. **b**–**d** Normalized frameshift products of different reporters in 293T (**b**),

U-2 OS (**c**) and SH-SY5Y (**d**) cells, respectively. All constructs were transiently transfected into cells for 24 h. NLuc signals were normalized to FLuc in each sample. The no-repeat frameshift is normalized by no-repeat control, and the repeat frameshift is normalized by the readthrough control. Data were reported as mean ± SEM from three or four biological replicates. Statistical analysis using two-tailed t-test. 293T: frame+1, ****$P = 0.000016$; frame+2, ****$P = 0.00007$. U-2 OS: frame+1, ****$P = 0.000021$; frame+2, ***$P = 0.00014$. SH-SY5Y: frame+1, ****$P = 0.000019$; frame+2, **$P = 0.0059$.

---

reporter contains an NLuc sequence in place of repeats, which has a similar length and indicates the elongation through a regular protein sequence.

To measure the elongation speed, we performed a ribosome runoff experiment in live cells[56,73]. Harringtonine, a translation inhibitor that prevents the first peptide bond formation, effectively stops translation initiation but does not interfere with existing 80S ribosomes on mRNAs[74]. Application of harringtonine synchronously stops initiation while allowing initiated ribosomes to run off the mRNA, the duration of which depends on the elongation speed and the length of the open reading frame (ORF)[75] (Fig. 4b). To facilitate long-term tracking of single mRNAs, the reporter mRNAs were tethered to the plasma membrane via membrane-localized MCP[73]. As expected, harringtonine results in a steady loss of translation intensity in the GA frame but with a slower speed for GP and GR (Fig. 4c–e, Supplementary Movie 7). We calculated the survival probability of the translation signal: the percentage of translating mRNAs as a function of time after adding Harringtonine ("Methods", only translating mRNAs at time zero were considered). The survival curve depends on the elongation of ribosome through the ORF: the curve shifts to the right if elongation is lower. Interestingly, it took much longer for the translation signal to disappear in the GP and GR frames than in the GA frame, which resembled the control NLuc reporter (Fig. 4f). The median runoff time for the GA reporter was 5.1 min, close to NLuc (4.9 min), whereas it was 11.2 min for GP and 13.5 min for GR (Fig. 4f), representing 2.3 and 2.8

folds increase respectively. Note that the GGGGCC repeat sequence (420 bp) only occupied a small fraction of the whole ORF (3009 bp) (Fig. 4g). The actual elongation speed discrepancy between DPRs was much larger than the difference in runoff time. To estimate the elongation speed through the repeat sequence, we calculated the ribosome residence time per amino acid in the repeat region, which was inversely proportional to the elongation speed (Fig. 4g, "Methods"). We estimated that the elongation speeds in the GP and GR frames were 10 and 15 folds slower compared with NLuc, while that of GA was statistically identical to NLuc (Fig. 4h).

We used fluorescence recovery after photobleaching (FRAP) of single translation sites as an independent method to measure elongation speed[58]. In agreement with the ribosome runoff experiment, we observed the TLS fluorescence in the control NLuc and GA reporters recovered rapidly, whereas the recovery was significantly slower in GP and GR reporters (Supplementary Fig. 5a, b, Supplementary Movie 8). Altogether, these results demonstrate that the GP and GR frames have slower elongation speed than the GA frame for the GGGGCC repeats.

### The amino acid composition of DPRs affects the translation elongation speed

The potential secondary structures formed by GGGGCC repeat RNA cannot explain the slow elongation through GP and GR since the GA frame shares the same secondary structure but elongates similarly with regular control sequences. We therefore investigated the

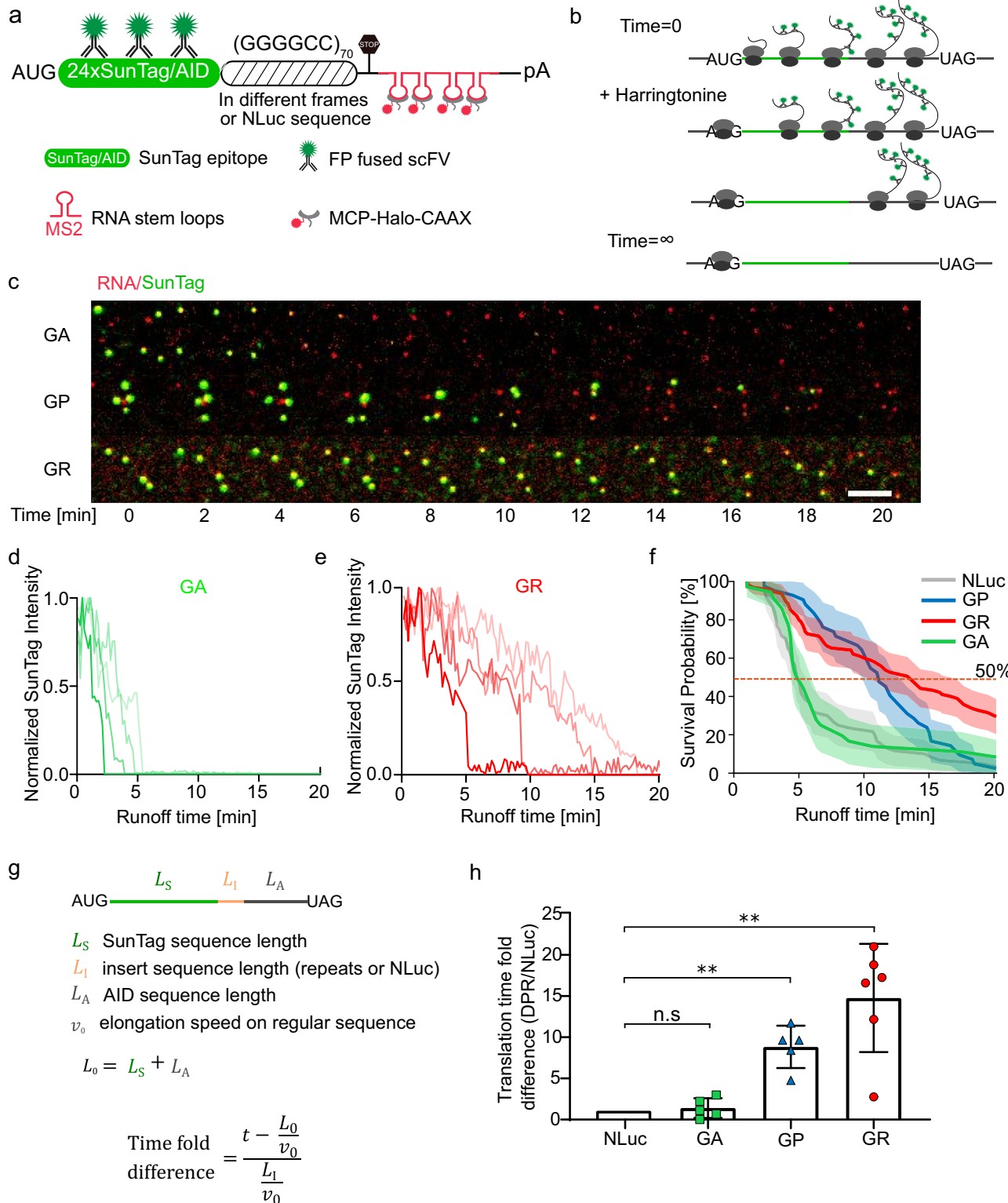

influence of the encoded amino acids. Proline introduces steric hindrance in the peptide and is known to cause ribosome pausing[76,77]. The antisense CCCCGG repeats also code for GP in one of the frames. Similarly, we observed a slower ribosome runoff in GP frame encoded by the antisense repeats (Supplementary Fig. 5a, c). The FRAP experiment also showed consistent results (Supplementary Fig. 5b). The amino acid arginine is positively charged. A stretch of arginine is a strong ribosome stalling sequence and induces ribosome quality

control in yeast[78]. Poly-GR has been shown to block the peptide exit tunnel of the ribosome in vitro[79]. We used synonymously randomized codons to encode poly-GR sequences. Similarly, we observed that the ribosome runoff was significantly slower than the NLuc control and similar to the repeat-encoded poly-GR (Supplementary Fig. 5c). Altogether, these results support that the amino acid composition of DPRs affects the translation elongation speed through the expanded repeat RNA.

**Fig. 4 | The translation elongation speeds in the GP and GR frames are slower than in the GA frame. a** Diagram of the single-molecule DPR elongation reporter construct. The (GGGGCC)$_{70}$ repeats were inserted after AUG initiated SunTag in different reading frames separately. The N-terminal SunTag fluorescence stayed on the RNA while the ribosome translated the repeats, therefore reporting the elongation speed. 24× MBS were used for RNA visualization and tethering to the plasma membrane through membrane-localized MCP (stdMCP-HaloTag-CAAX). **b** Schematics of ribosome runoff assay. Harringtonine effectively stops translation initiation but does not affect elongating ribosomes, which continue translating until "runoff". This assay is used to determine the elongation speed. **c** Snapshots from videos of ribosome runoff experiment for GA, GP and GR reporter cells, respectively. The time reported was after adding harringtonine (Supplementary Movie 7). Scale bar: 5 μm. **d, e** Example translation intensity traces for GA (**d**) and GR (**e**) reporters after harringtonine treatment. Traces were normalized to the respective maximum intensities. **f** The survival curves of translation sites as a

function of runoff time ("Methods"): the percent of mRNAs still having translation signals at given times was calculated in each cell and then averaged. Dashed line represents the 50% of mRNAs that finished runoff. The shadow represents the 95% confidence bounds (Greenwood's formula). NLuc: 6 cells, 60 TLS; GP: 5 cells, 42 TLS; GR: 6 cells, 95 TLS; GA: 5 cells, 36 TLS. **g** Estimation of elongation speed through the inserted sequence using median runoff time measured in (**f**) ("Methods"). The time spent translating the DPR is compared with translating NLuc. $L_S$: length of SunTag; $L_A$: length of AID; $L_0 = L_S + L_A$ is identical for all reporters; $L_I$: length of inserted sequence (NLuc or repeats); $v_0$: regular elongation speed (through NLuc or the rest reporter); t: runoff time. **h** Elongation time (per amino acid) through different repeat frames compared with through NLuc (median). Each symbol represents the average measurement in one cell. Data are mean ± SD from six (Nluc, GR) or five (GP, GA) independent experiments. Statistical analysis using one-sample t-test: GA, $P = 0.50$; GP, $**P = 0.0024$; GR, $**P = 0.0037$.

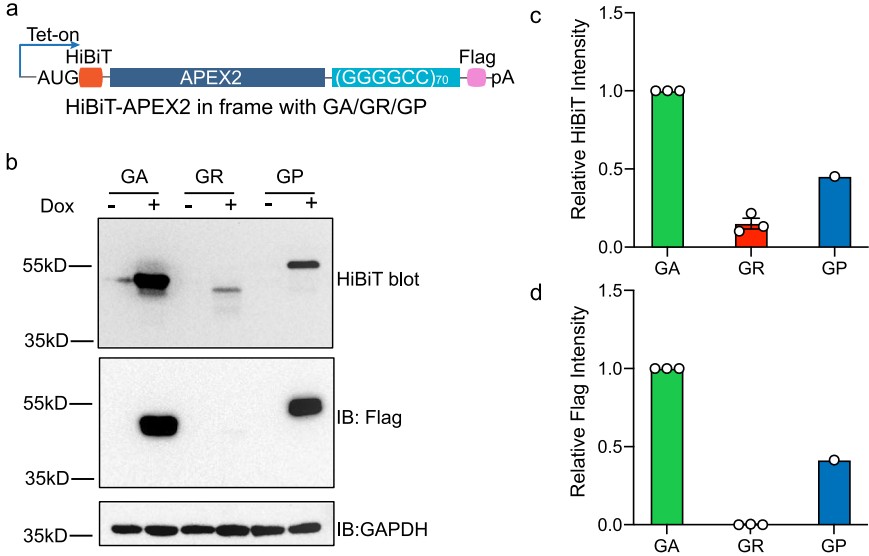

**Fig. 5 | The production of poly-GR from AUG-translation is low. a** Diagram of DPR elongation luciferase reporters. The DPR of interest is in frame with N-terminal AUG-HiBiT-APEX2 and C-terminal Flag tag. **b** HiBiT blotting (top) or western blotting with Flag antibody (bottom) to measure the DPR products with N- and C-terminal tags. **c** Densitometry quantification of the relative reporter expressions by HiBiT blot. Data are presented as mean ± SEM from three (GA, GR) or one (GP) independent experiments. **d** Densitometry quantification of the relative reporter expressions by Flag blot. Data are presented as mean ± SEM from three (GA, GR) or one (GP) independent experiments.

To assess the effect of elongation on the final protein product, we designed a split luciferase reporter to measure DPR production. The AUG-driven 11-amino acid fragment of NanoLuc (HiBiT) was fused upstream of the GGGGCC repeat in different frames to measure the N-terminal product. A Flag tag was placed at the C-terminus in frame with AUG-HiBiT to report the full-length proteins (Fig. 5a). The construct was controlled by an inducible Tet-On promoter and stably integrated through the Flp-In system in HeLa cells. The protein level was measured through HiBiT blotting or immunoblotting against Flag. When the GGGGCC repeat was placed in the GA frame, the protein level was significantly higher than that in the GP and GR frames (Fig. 5b–d) after the RNA level was normalized (Supplementary Fig. 6a). As all the DPR translation was initiated by AUG start codon in the same reporter backbone, the discrepancy cannot be explained by the initiation differences. We next tested whether the protein stability contributes to the different steady-state DPR levels. We measured the DPR stability and found that poly-GA was, in fact, less stable than poly-GP or poly-GR (Supplementary Fig. 6b–f). Therefore, the higher poly-GA expression level was not due to its increased stability either. Taken together, we conclude that the

protein amount correlates with the translation elongation rate in different DPR frames.

To understand the potential mechanism of how elongation may influence the protein yield, we performed in vitro translation of the elongation reporters (Supplementary Fig. 6g). The in vitro transcribed reporter mRNAs (Supplementary Fig. 6h) were capped and translated in rabbit reticulocyte lysate. We included the K20 (K$^{AAA}$)$_{20}$ reporter as a positive control, which is known to induce translation stalling[80]. The no repeat construct (K0) was used as negative control for stalling. As expected, there was barely any detectable translation product from the K20 reporter RNA (Supplementary Fig. 6i). The translation yield from the GA reporter was comparable to the no repeat control (Supplementary Fig. 6i), consistent with the in vivo translation measurement that the elongation in GA frame was similar to the NLuc. Interestingly, we observed less full-length protein from the (GGGGCC)$_{40}$-GR reporter. There were multiple shorter protein bands, likely representing truncated translation products (Supplementary Fig. 6i). This indicates that there is ribosome stalling and truncation of the nascent peptide during translation elongation in the GR

frame, which could contribute to reduced poly-GR production compared to poly-GA.

### RQC factors modulate the translation dynamics of DPRs

To identify genetic modifiers for poly-GR production, we performed a CRISPR-Cas9 knockout screening. We constructed an AUG-driven translation reporter of GGGGCC repeat in the GR frame fused with a GFP at the C-terminus. We stably expressed it in the HeLa Flp-In cells under the Tet-On promoter (Fig. 6a). The reporter cells were infected with a lentiviral guide RNA (gRNA) library targeting genes encoding 1906 RNA-binding proteins and ribosome proteins. After drug selection for gRNA expression, GR-GFP was induced by doxycycline. The GFP-high and GFP-low cell populations were collected by fluorescence-activated cell sorting (FACS) (Supplementary Fig. 7a, b)[81]. We sequenced the gRNA fragments from genomic DNA in each cell population and analyzed their distribution[81]. Many protein/RNA quality control genes, such as *ZNF598* and *Pelota*, were top hits in the screen whose gRNAs were enriched in the GFP-high population, suggesting knockdown of these genes can increase GR-GFP level (Fig. 6b, Supplementary Data 1).

We further used the single-molecule imaging platform to directly evaluate the function of RQC factors on DPR translation. The slow-down or stalling of ribosomes may lead to the collision of trailing ribosomes. The collided ribosomes signal aberrant translation status, resulting in the splitting of ribosomes and subsequent degradation of nascent peptides and mRNAs, known as the RQC pathway[82,83]. ZNF598 is an E3 ubiquitin ligase that detects the collided ribosomes, ubiquitinates the small ribosomal subunit and triggers RQC response[84,85]. In the polyA-induced ribosome collision, ZNF598 accelerated the clearance of the collided ribosomal queue[73].

As the elongation speed through poly-GR is significantly slower than a general protein sequence, it may cause trailing ribosomes to collide. That might be the reason why many RQC factors were enriched in the CRISPR screen. To test the hypothesis, we evaluated whether ZNF598 knockdown influenced the translation kinetics of the elongation reporters (Fig. 4a). We performed ribosome runoff assays as described previously and compared the ZNF598 siRNA transfected cells with scrambled siRNA control (Supplementary Fig. 7c, d). Knocking down ZNF598 further extended the ribosome runoff time of the (GGGGCC)n-encoded GR (Fig. 6c, d, Supplementary Movie 9), but interestingly had no effect on the GR reporter encoded by randomly synonymized codons (Fig. 6e). More surprisingly, the ZNF598 knockdown also slowed down the ribosome runoff on the (GGGGCC)n-encoded GA repeat (Fig. 6f), even though its elongation speed was similar to the control NLuc sequences (Fig. 4f). Likewise, ZNF598 knockdown had no effect on randomized GA (Fig. 6g). *Pelota* is the paralogue of translation release factor *eRF1*[86]. The yeast homolog of *Pelota* is implicated in recycling ribosomes at the truncated message[49]. Knocking down Pelota recapitulated the observation from the ZNF598 knockdown experiment (Fig. 6d–g, Supplementary Fig. 7e). Altogether, our data support that the translation of C9ORF72 repeats can be modulated by the ribosome-associated quality control pathway. The repeat RNA sequence, rather than the DPR composition, is important to be recognized as the RQC substrate.

To evaluate whether RQC factors were involved in diseases, we further analyzed the proteomics data of C9ORF72-ALS/FTD patient-derived induced pluripotent stem cell (iPSC)-differentiated neurons (iPSN) from AnswerALS[87]. C9ORF72-ALS patient neurons showed a significantly decreased level of Pelota protein compared to non-neurological controls (Fig. 6h). This could influence the DPR production as well as other protein homeostasis in neurons, potentially contributing to the disease pathogenesis.

## Discussion

Despite recent progress in understanding the RAN translation initiation mechanisms in C9ORF72-ALS/FTD, little is known about translation dynamics through the expanded repeats. Previous work mostly relied on ensemble approaches to measure the final DPR level, which was an indirect readout of translation activity. In this study, we used single-molecule imaging in live cells to directly examine the translation kinetics and investigated various aspects of repeat-associated translation: initiation, elongation, frameshift, and quality control. We recently identified that the spliced intron of *C9ORF72* was stabilized by the GGGGCC repeats in a circular form, which was exported to the cytoplasm and subjected to cap-independent translation[72]. In this paper, the repeat was placed in a capped mRNA reporter for translation analysis, as this allowed a higher expression level and more accurate quantification of translation properties. We used the canonical AUG start codon for translation initiation when assessing the elongation, frameshift, and quality control response in different DPR frames, which was independent of the initiation step.

To study RAN translation, we and others have placed reporters after the GGGGCC repeats in different frames to report the in-frame DPR production separately[36,37]. With the limitation in mind, we found that in agreement with previous work[37,38], translation initiation in the GA frame had higher efficiency than GP and GR frames, and the CUG near-cognate start codon proceeding the repeats contributed to the initiation in the GA frame. However, the difference in the initiation rate was insufficient to explain the measured difference in the final DPR amounts. Instead, we found that the elongation speeds of different reading frames varied significantly, with GP slower than GA and GR the slowest. The discrepancy of translation elongation may contribute to the different amounts of the final DPR products. It is noted that this is not direct proof that the specific DPR is produced from the initiation of the labeled frame because frameshift and abortive translation can occur. Nevertheless, we have shown that frameshift is rare. On the first-order approximation, we believe that the initiation reporters can represent the RAN translation efficiency.

We directly observed frameshift between different frames with a two-color translation assay. The frameshift frequency on the repeats was higher than the regular RNA sequences. Therefore, chimeric DPRs likely exist in patients. However, the frequency was overall low and unlikely to be the major determinant for the DPR abundance. Contrary to the previous hypothesis, we found that the frameshift from GR to GA had a higher probability than the reverse. This was likely due to the slow elongation speed of GR, which could increase the chance of frameshift. Furthermore, mutating the near-cognate CUG codon in the GA frame did not reduce the translation efficiency of GP and GR frames. Taken together, our study supports that the majority of poly-GP and poly-GR should arise from independent initiation in the corresponding frames rather than frameshift from poly-GA. It is possible that longer expanded repeats in patients can slightly increase the chance of frameshifting, which cannot be fully recapitulated by the 70× repeat reporter used in this study. Nevertheless, the relative frequency among different frameshifting directions (such as higher GR-to-GA than GA-to-GR) should remain similar. Consistent with our finding, a recent study demonstrated that homozygous deletion of an intron fragment preceding the endogenous repeats, which contains the CUG codon, in C9ORF72-ALS/FTD patient-derived iPSNs only reduced the level of endogenous poly-GA but not poly-GP and poly-GR[88]. This evidence strongly suggests that frameshifting is not the main mechanism for poly-GP or poly-GR production from the GGGGCC repeats. Overall, our work indicates that there are chimeric DPRs produced via translation frameshift in C9ORF72-ALS/FTD patients, but the proportion of chimeric peptides could be low.

The GGGGCC repeat is prone to form strong secondary structures and may slow down elongation. However, we showed that the coded

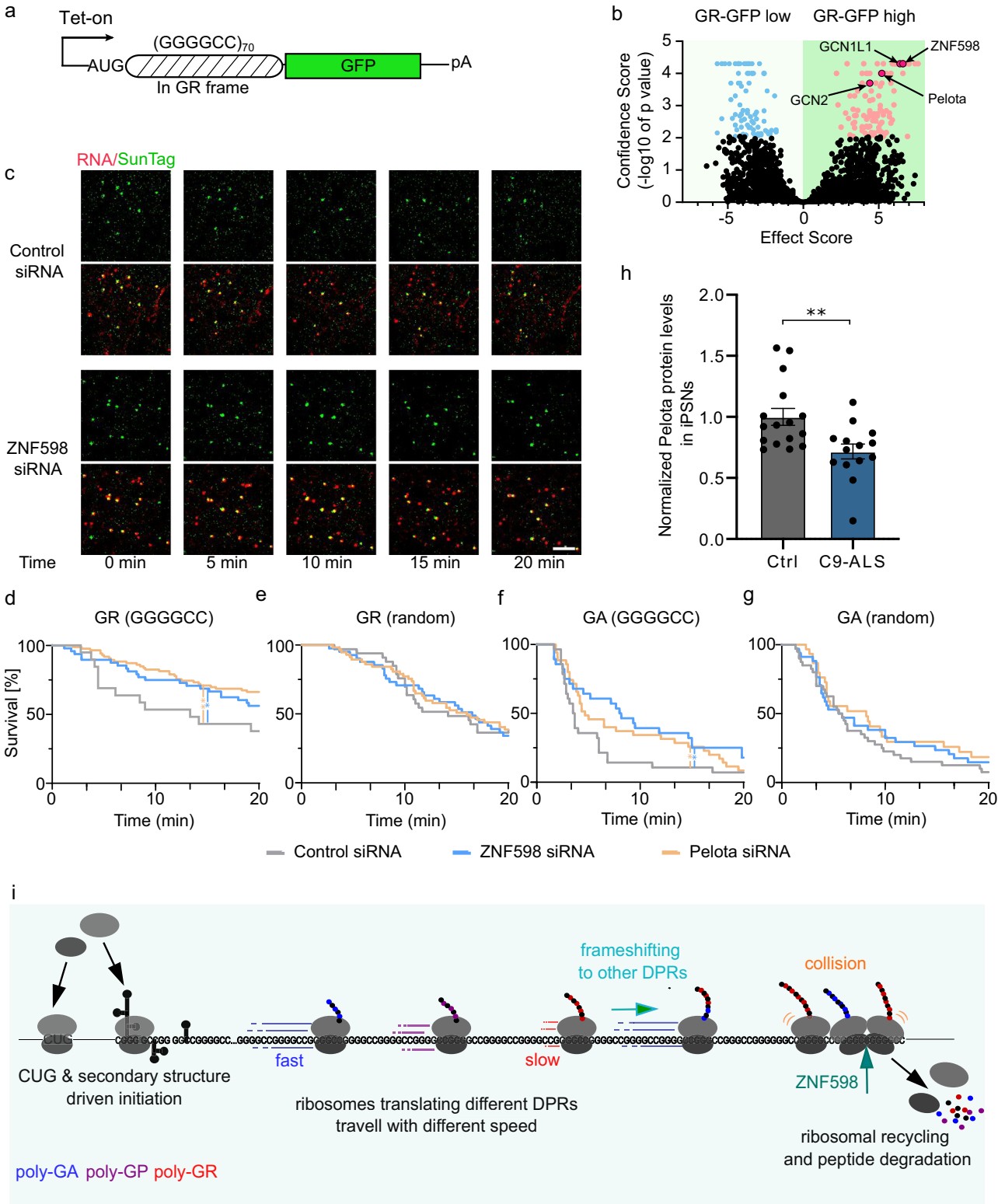

amino acid, rather than the secondary structure of the RNA molecule, was the main determinant for the different elongation speeds between frames. The GA frame had a similar elongation speed as a general protein, but the GP and GR frames had much slower elongation, even though the RNA template was the same. The slow elongation of GP and GR frames was unlikely due to the rare codon effect because the antisense (CCCCGG)n- or the randomly synonymized poly-GP or poly-GR had similar elongation speeds compared to the (GGGGCC)n-

encoded ones. Therefore, the amino acid composition of the DPRs, rather than RNA or codon sequences, determines the elongation speed. Recently it has been shown that the positively charged poly-GR interacted with the peptide exit tunnel[79], a possible mechanism for its slow elongation. Additionally, the bulky proline residue may cause a steric hindrance and slow down the elongating ribosomes[76,79,89].

Aberrant mRNAs can lead to ribosomal stalling and degradation of nascent proteins and/or mRNAs[90,91]. Cells can readily resolve the

**Fig. 6 | Ribosome quality control factors influence elongation dynamics through repeat RNAs. a** Diagram of doxycycline-inducible AUG-driven GR-GFP translation reporter for CRISPR screening. The reporter was stable integrated in Hela cells. **b** Volcano plot depicts gene knockout enrichment in cells with different GR levels. The top GFP-high and GFP-low cell populations were collected by FACS and subjected to deep sequencing and statistical analysis for gRNA distribution. Red: genes contributing to high GR-GFP levels when knocked out (10% FDR), Blue: genes contributing to low GR-GFP levels when knocked out (10% FDR). **c** Representative time-lapse images of runoff experiment for scrambled control siRNA (top) and ZNF598 siRNA transfected GR (GGGGCC)$_{70}$ elongation reporter cells (Fig. 4a). Scale bar: 2.5 μm. Also see Supplementary Movie 9. **d–g** The translation site survival curves of ribosome runoff experiment for control (gray), ZNF598 knockdown (blue), and Pelota knockdown (orange) in elongation reporter cells: GR-(GGGGCC)$_{70}$ (**d**), GR random (**e**), GA (GGGGCC)$_{70}$ (**f**) and GA random (**g**). For DPR random reporters, the repeat sequence was replaced with randomly

synonymized sequences coding for the same DPR of interest. The graphs were generated from 4–6 cells per condition, with 48 TLS in GR(GGGGCC), 41 TLS in GR(random), 28 TLS in GA(GGGGCC), 34 TLS in GA (random), for ZNF598 siRNA KD; 86 TLS in GR(GGGGCC), 57 TLS in GR(random), 35 TLS in GA(GGGGCC), 30 TLS in GA(random)for Pelota siRNA KD and with 19 TLS in GR(GGGGCC), 33 TLS in GR(random), 28 TLS in GA(GGGGCC), 40 TLS in GA(random) for Control siRNA KD. The survival curve comparison between Control and ZNF598 and Pelo groups was done using Log-rank (Mantel-Cox) test. GR(GGGGCC): ZNF598 siRNA, *$P = 0.042$; Pelota siRNA, **$P = 0.0051$. GA(GGGGCC): ZNF598 siRNA, *$P = 0.016$; Pelota siRNA, *$P = 0.042$. Data are median survival. **h** Normalized Pelota protein levels in iPSN derived from control ($n = 16$) and C9-ALS patients ($n = 14$). Graph shows proteomics measurements from AnswerALS dataset[87]. Data are mean ± SEM. Statistical test using two-tailed nonparametric Wilcox test: **$P = 0.0045$. **i** A working model showing the influence of translation initiation, elongation, frameshift and quality control factors on the DPR productions.

transient ribosome pause, but prolonged stalling will invoke RQC[73]. Slow elongation may induce ribosome collision and associated quality control. We found that the RQC factors ZNF598 and Pelota modulate translation dynamics through repeats. Surprisingly, both poly-GA and poly-GR encoded by GGGGCC repeats were modulated, but not the DPR encoded by synonymously randomized codons. This indicates that the repeat RNA sequence or structure, instead of the DPR amino acid composition or elongation rate, is important for invoking the quality control mechanism. There is precedence for this phenomenon: poly-lysine encoded by AAA was more efficient to induce RQC than the same peptide encoded by AAG[92]. It is worth further study on how GGGGCC structurally interacts with collided ribosomes to recruit RQC factors. In C9ALS patient-derived iPSN lines, Pelota protein level is lower, which could lead to higher DPR expression and exacerbate the disease. Mutation in *ZNF598* (R69C) has been linked to ALS from large-scale genome-wide association study[93]. The dysregulation in RQC pathways will be a risk factor not only for increasing DPR expression but also for generating global cellular stress[94].

The ribosome collision and the queue formed by stalling sequence depend on the initiation rate and repeat length. A higher initiation rate may lead to more frequent collisions and cause a stronger RQC effect. The elongation reporters in this study were driven by canonical AUG, with an initiation rate much higher than RAN translation. The low ribosomal flux for RAN translation may reduce the collision rate or allow the collision to be naturally resolved and bypass detection. On the other hand, longer repeats in patients may have more translating ribosomes or take a longer time to translate through. In addition, the RAN translation may occur in any frame of the repeat. If a ribosome stalls in one frame (such as GR), it will stop ribosomes in any frame, including GA, which has a higher initiation and elongation rate. In this work, we only used 70x GGGGCC repeats due to technical difficulty to construct longer repeats. Longer repeat in patient may lead to higher levels of RQC.

The RNA-mediated gain-of-toxicity in C9ORF72-ALS/FTD may result from the formation of RNA granules that sequester RNA-binding proteins (RBP) and disrupt RNA metabolism[95]. The RBPs play essential roles in regulating RNA splicing, export, stability as well as translation. It is plausible that some RBPs may influence translation initiation, elongation, termination or frameshift, for example, by influencing the repeat RNA structures. However, it is unlikely that RBPs may exert differential influences on different DPR frames. In addition, the sub-cellular localization of the RBPs should be considered for their pathophysiological functions. The repeat RNA granules are predominantly in the nucleus, while the translation occurs in the cytoplasm. The single-molecule imaging systems reported here may provide a valuable platform to study how specific RBPs can affect the different properties of repeat RNA translation.

DPR level is regulated by diverse cellular signaling pathways. C9ORF72 DPR production is upregulated by the integrated stress response through eIF2α phosphorylation[36,37]. Increased neuronal activity can elevate the DPR levels[96]. The protein kinase A (PKA) pathway has been shown to positively influence RAN translation[97]. It is generally believed that the repeat translation initiation is modulated by these signaling pathways. It will be interesting to explore whether they also influence the translation elongation or frameshifting of the expanded repeats.

Taken together, this work studied various aspects of translation associated with GGGGCC repeat in C9ORF72-ALS/FTD. The single-molecule approach points out the importance of direct measurements of multiple translation steps to achieve a better understanding of the repeat RNA translation. This platform can be applied to other repeat expansion diseases. Our study reveals that the final DPR level depends on a multifaceted mechanism (Fig. 6i), including initiation, elongation, frameshifting, and quality control pathways. This work also sheds light on other repeat expansion diseases that it is important to examine different translation steps and properties besides the initiation. Different RNA or amino acid repeat sequences probably have different influences on translation, which can be explored using similar single-molecule imaging approaches. Understanding the mechanism of the repeat RNA translation and DPR production is of critical importance to identify the genetic modifiers and create a therapeutic intervention.

## Methods

### Plasmid

**Reporters to measure translation initiation.** To compare the RAN translation efficiency between different frames, pCAG-(GGGGCC)$_{70}$-24×SunTag-24×MBS mRNA expression plasmids were constructed. 24×MBSV5 was PCR amplified from puc57-24×MBSV5 template[63] and inserted downstream of (GGGGCC)$_{70}$ via NsiI and SacI in the pCAG-(GGGGCC)$_{70}$ vector[81]. The FLAG-24×SuntagV4 was cut out from pUbC-FLAG-24×SuntagV4-oxEBFP-AID-baUTR1-24×MBSV5-Wpre[58] (Addgene Plasmid #84561) by NotI and FspI and inserted into pCAG-(GGGGCC)$_{70}$-24×MBS via NotI and NsiI (Blunt) between (GGGGCC)$_{70}$ and 24×MBS. The ATG start codon was removed, and a frameshift was introduced by replacing the region between NotI and AflII with a linker sequence AGATTACAAGGACGACGACGATAAGGGCGGACCGGGTGGATCTGGAGGTGGAGGTTCTGGAGGAGAAGAACTTTTGAGCAAGAATTATCATCTTGAGAACGAAGTGGCTCGT for GA frame, GATTACAAGGACGACGACGATAAGGGCGGACCGGGTGGATCTGGAGGTGGAGGTTCTGGAGGAGAAGAACTTTTGAGCAAGAATTATCATCTTGAGAACGAAGTGGCTCGT for GR frame, and TCGATTACAAGGACGACGACGATAAGGGCGGACCGGGTGGATCTGGAGGTGGAGGTTCTGGAGGAGAAGAACTTTTGAGCAAGAATTATCATCTTGAGAACGAAGTGGCTCGT for GP frame. For reporters with CUG to CGG mutation, the preceding intron region was replaced by (GTGAACAAGAAAAGACCTGATAAAGATTAACCAGAAGAAAACAAGGAGGGAAACAACCGCAGCCTGTAGCAAGCTCGgGGAACTCAGGAGTC). For the pCAG-stop×3-24×SunTag-

24×MBS no repeat control plasmid, the (GGGGCC)$_{70}$ repeat sequence between HindIII and NotI (blunt) was replaced by the intron sequence cut from the previous bicistronic plasmid (Cheng 2018, Nat Comm) by HindIII and BssHII (blunt).

To make bicistronic splicing luciferase reporters with CGG mutation, the preceding intron region between HindIII and BSSHII of our previous bicistronic splicing reporter[66] was replaced by DNA sequences containing CGG mutation.

**Reporters to measure translation elongation.** For elongation reporters, ATG-SunTag was placed in front of repeat sequences. To make pCAG-ATG-24×SunTag-MCS-24×MBS plasmids, BFP sequence after SunTag in pCAG-ATG-24×SunTag-24×MBS was replaced with fragments containing multiple cloning sites (MCS) using PstI and BsrGI: MCS1 (AAGCTTGCAGGTACCGGAGCAGATATCGCTGGAGCGG CCGC) for (GGGGCC)$_{70}$ GA frame and (CCCCGG)$_{70}$ GP frame; MCS2 (AAAGCTTGCAGGTACCGGAGCAGATATCGCTGGAGCGGCCGCAT) for (GGGGCC)$_{70}$ GR frame. MCS3 (AAAAGCTTGCAGGTACCGGAGC AGATATCGCTGGAGCGGCCGCT) for (GGGGCC)$_{70}$ GP frame. Next, the (GGGGCC)$_{70}$ repeat was cut from PBS-(GGGGCC)$_{70}$ plasmid[66] by KpnI and XhoI (DNA blunting at the XhoI end) and cloned into pCAG-ATG-24×SunTag-MCS-24×MBS plasmids at the KpnI + EcoRV sites. To make antisense GP frame reporter, pBS-(GGGGCC)$_{70}$ was cut by HindIII and Acc65I (blunt) and inserted into pCAG-ATG-24×SunTag-MCS1-24×MBS by HindIII and EcoRV. To make control plasmid, NLuc sequence (513 bp) was PCR amplified and inserted into pCAG-ATG-24×SunTag-MCS1-24×MBS by HindIII and EcoRV. For elongation reporters with randomized codon, the DNA fragments encoding GA50, GP50 and GR50 were synthesized (Genewiz) and ligated to produce GA100, GP80 (shortened during cloning) and GR100. GP80 was cloned into pCAG-ATG-24×SunTag-MCS2-24×MBS by EcoRV and NotI. GA100 and GR100 cloned into pCAG-ATG-24×SunTag-MCS1-24×MBS by HindIII and EcoRV.

**Reporters to measure translation frameshift.** For frameshift reporters, a fragment containing multiple cloning sites (CCTGCAGGGAA TTCTTAATTAAATGCCTCGAGACTAGTTGTACATAGAAGCTTGCCCGG GCGCGCGCGGATCCGATATC) was first placed between AflII and XbaI in pcDNA5-TO plasmid. 24×rHA-sNluc-smAID was synthesized (Genewiz) and cloned into pcDNA5-TO-MCS by XhoI (blunt) and BsrGI. Next, (GGGGCC)$_{70}$-24×SunTag-24×MBS was cut from pCAG-Δintron-(GGGGCC)$_{70}$-24×SunTag-24×MBS (GA frame) by NotI (blunt) and BglII and inserted into the plasmid at the SrfI and BamHI sites. The 24×MBS was replaced with 24×MBS-12×PBS cut from pcDNA5-ATF4-24×SunTag-24×MBS-12×PBS[73] by NheI and NsiI (blunt) to make the final pcDNA5-TO-24×rHA-(GGGGCC)70-24×SunTag-24×MBS-12×PBS reporter (GR to GA reporter). For the no repeat control reporter (pcDNA5-TO-24×rHA-stop×3-24×SunTag-24×MBS-12×PBS), the (GGGGCC)$_{70}$-24×SunTag in the GR to GA reporter was replaced with 24×SunTag from pCAG-stop×3-24×SunTag-24×MBS by HindIII and NheI. For the dual-color GA RAN reporter (pcDNA5-TO-24×rHA-stop×3-(GGGGCC) 70-24×SunTag-24×MBS-12×PBS), the (GGGGCC)$_{70}$-24×SunTag was cut from pCAG-Δintron-(GGGGCC)$_{70}$-24×SunTag-24×MBS (GA frame) and inserted into the above no repeat control plasmid by NotI and NheI. The stop codons in all three reading frames were subsequently placed between HindIII and NotI. To make HA-GA-SunTag readthrough reporter, the stop codon in GA frame between HindIII and NotI was removed to make HA in frame with GA reading frame of (GGGGCC)$_{70}$. For frameshift GA to GR reporter, (GGGGCC)$_{70}$-24×SunTag (GR frame) was cut from pCAG-Δintron-(GGGGCC)$_{70}$-24×SunTag-24×MBS (GR frame) and inserted into HA-GA-SunTag readthrough reporter by NotI and NheI. Similarly, to make GA to GP reporter, (GGGGCC)$_{70}$-24×SunTag (GP frame) was cut from pCAG-Δintron-(GGGGCC)$_{70}$-24×SunTag-24×MBS (GP frame) and inserted into HA-GA-SunTag readthrough reporter by NotI and NheI.

**Luciferase reporters for frameshift.** For dual-luciferase frameshift reporters, FLuc and NLuc were PCR amplified from template. To keep NLuc in frame, in +1 frame and in +2 frame with FLuc, one or two nucleotides were introduced into the linker sequence GCTTGCGGCCGCCTAGGTAGC, GCTTGCGGCCGCtCTAGGTAGC, GC TTGCGGCCGCttCTAGGTAGC, respectively. pcDNA5-FRT-TO vector was digested by HindIII/NotI. The FLuc, NLuc (0, +1, +2 frame) and the digested vector were ligated together via Gibson assembly. Then the (GGGGCC)$_{70}$ repeat was inserted into the vector via BssHII and NotI.

**Reporters to measure the stability of different DPRs.** To study the stability of different DPRs, DNA fragments encoding GA50, GP50 and GR50 with randomized codon were synthesized (Genewiz). The HiBiT tag with ATG start codon and 2× Flag were introduced to the N-terminal and C-terminal of those DNA fragments, respectively. Next, the whole fragment was cloned into DOX inducible lentiviral vector by NheI and AgeI to generate Lenti-TRE3G-HiBiT-GA50/GP50/GR50-2×Flag.

**AUG-GGGGCC$_{70}$ (GR)-EGFP reporter.** To construct GR-EGFP reporter for CRISPR screening, HiBiT-HA tag with ATG start codon (AAGCTTACCATGGTGAGCGGCTGGCGGCTGTTCAAGAA-GATTAGCGGTTCAAGTGGATACCCATACGACGTCCCA-GACTACGCTGGGTAC) was introduced into the 5' end of EGFP by PCR amplification. The fragment was then inserted into pcDNA5-FRT-TO vector via HindIII and NsiI. Next, (GGGGCC)$_{70}$ repeats were placed between HA tag and EGFP by KpnI and XhoI.

**In vitro translation reporter.** The K0 and K20 reporters were constructed by PCR amplification from pmGFP-P2A-K0-P2A-RFP and pmGFP-P2A-K20-P2A-RFP (Addgene plasmid # 105686 and # 105688)[80] with N-terminal HiBiT-HA tag and a C-terminal 2× Flag tag. The amplified sequence was inserted into pBluescript II SK(+) by HindIII and NotI. The GGGGCC$_{40}$ fragment was cut from the pBS-GGGGCC$_{40}$ plasmid[81] via KpnI and EcoRV and cloned into the same sites of the above K0 reporter to produce the GGGGCC$_{40}$(GA) reporter. The GGGGCC$_{40}$ fragment was cloned into the K0 reporter at KpnI and SalI (blunt) sites to generate the GGGGCC$_{40}$(GR) reporter.

## Cell culture and transfection

U-2 OS (American Type Culture Collection HTB-96), HEK293T (American Type Culture Collection CRL-1573), HeLa (American Type Culture Collection CCL-2), HeLa Flp-In (Thermo Fisher), and SH-SY5Y (American Type Culture Collection CRL-2266) cells were cultured in DMEM supplemented with 10% (v/v) FBS, 100 U/mL penicillin, and 100 μg/mL streptomycin.

i³N iPSC[98] were grown in E8 medium (Thermo Fisher, A15169-01) on Matrigel (Corning, 354277) coated plates. For neuron differentiation, iPSC cells were first seeded and cultured in Neuronal Induction Medium (DMEM/F12 containing N-2 supplement, Nonessential amino acids, Gluta-MAX, 10 μM Y-27623 and 2 μg/mL doxycycline) for 3 days. Then cells were dissociated with Accutase (MilliporeSigma, A6964) and seeded on a 24-well plate pre-coated with 0.1 mg/mL poly-L-ornithine (Sigma, P3655) at a density of $3 \times 10^5$ per well. Cells were maintained in BrainPhys Neuronal Medium (STEMCELL Technologies), containing B-27 supplement, 10 ng/mL BDNF, 10 ng/mL neurotrophin-3, and 1 μg/mL laminin, with half-media changes every other day until harvest.

To generate stable cell lines, U-2 OS cells, U2PA stably expressing stdMCP-HaloTag and scFv-sfGFP for initiation reporters, U-2 OS stably expressing stdMCP-HaloTag-CAAX and scFv-sfGFP for initiation and elongation reporters (live cell imaging), U-2 OS stably expressing stdMCP-tagRFPT, stdPCP-CAAX, scFv-sfGFP (SunTag targeting), and Fb-HA-HaloTag (HA targeting for frameshift reporters)

were transfected with linearized reporter constructs by nucleofection (Lonza nucleofector 2b, VACA-1003), and followed by drug selection (200 µg/mL hygromycin or 1 µg/mL puromycin) for 3 weeks. HeLa Flp-In stable cell lines expressing bicistronic splicing reporters with CGG mutation were generated as described before[66]. For transient transfection experiments, TransIT®-LT1 Transfection Reagent (Mirus Bio) or X-tremeGene HP transfection reagent (Roche, 6366236001) was used for plasmid transfection, and Lipofectamine® RNAiMAX (Invitrogen) was used for siRNAs transfection. ON-TARGETplus pooled siRNAs of *ZNF598*, *PELO*, and nontargeting control (GE Dharmacon) were transfected twice at 0.25 nM, and cells were imaged 24 h later after the second transfection.

## Dual-luciferase assay

Nano-Glo Dual Luciferase Assay system (Promega, N1620) was used to measure the NLuc and FLuc luciferase activities in cells transfected with frameshift reporters according to the manufacturer's instructions. NLuc levels were normalized to FLuc in each sample for comparison. Three biological replicates were included at each condition.

## DPR stability measurement

To study the stability of different DPRs, Dox inducible lentiviral plasmids expressing HiBiT tagged GA50/GP50/GR50 were packaged into lentivirus. Neurons were infected on day 4 post-differentiation with 100% MOI. On day 6, DPR expression was induced by adding doxycycline (2 µg/mL). Doxycycline was removed from day 8, and cells were collected every day (including day 8) till day 14. Cells without doxycycline induction on day 6 were also collected on day 8 as a non-induction control. Then the HiBiT activity, which represents the levels of DPRs, was measured by the Nano-Glo® HiBiT Lytic Detection System (Promega, N3030) according to the manufacturer's instructions. To compare the relative protein level among different samples, the HiBiT activity was normalized to the total protein level measured by BCA assay (Thermo Fisher Scientific).

## In vitro transcription and translation

RNA was in vitro transcribed from NotI linearized vector using MEGAscript T7 Transcription Kit (Thermo Fisher Scientific, AM1333) in the presence of $m^7GpppG$ cap analog (NEB, S1404). Reactions were carried out at 37 °C for 3 h. Synthesized RNAs were purified by TRIzol reagent. RNA size and quality were verified by agarose gel. Then, 500 ng mRNAs were in vitro translated with Flexi Rabbit Reticulocyte Lysate System (Promega, L4540), following the manufacturer's protocol. The reactions were incubated at 30 °C for 30 min before termination by incubation at 4 °C. Next, 10 µL reaction was mixed with 40 µL 2x Laemmli sample buffer and heated at 70 °C for 15 min, then 20 µL was loaded on 15% polyacrylamide gel. Expression was detected by Nano-Glo® HiBiT Blotting detection kit (Promega, PRN2410).

## Immunoblotting

For immunoblotting, goat anti-mouse or anti-rabbit IgG HRP-conjugated (1:5000, GE healthcare, #NA934V and #NA934V) was used along with chemiluminescent detection reagents (Thermo Scientific). The primary antibody included GAPDH (1:1000, Cell Signaling Technology, #2118), FLAG (1:1000, Sigma, #F1804), ZNF598 (1:1000, GeneTex, #GTX119246), PELO (1:1000, proteintech, #10582-1-AP).For HiBiT tag detecting, Nano-Glo® HiBiT Blotting System (Promega, N2410) was used according to the manufacturer's instructions. Uncropped blots for Fig. 5b and Supplementary Fig. 7d, e can be found in the Source Data file.

## CRISPR screening

Guide RNAs (gRNAs) targeting genes encoding 1906 potential RNA-binding proteins (RBPs), ribosome proteins and nontargeting negative control gRNAs were chosen from the genome-wide guide RNA library to make RBP sub-library as described previously[99]. The RBP sub-library was packaged into lentivirus using 293T cells. About 30 million HeLa Flp-In cells expressing AUG-$(GGGGCC)_{70}$-EGFP (GR in frame with AUG start codon and EGFP) were infected with the library at an MOI of 0.4. After infection, cells were placed under puromycin selection at 1 µg/mL for 3 days. After drug selection, 2 µg/mL doxycycline was applied to cells to induce the reporter expression. Cells were split every 3 days or as needed. Throughout the screen, the cells were maintained at ≥1000× gRNA coverage to keep the library complexity. After 14 days of culture, 20 million cells were analyzed by fluorescence-activated cell sorting (FACS), and the top 5% GFP-high and 5% GFP-low cell populations were collected[81]. Genomic DNA was extracted from the individual cell population using the QIAamp Blood Mini Kit (Qiagen). The DNA fragment libraries containing gRNAs were generated by PCR and subjected to high-throughput sequencing on the Illumina Nextseq platform. The enrichment of individual gRNAs in GFP-high vs GPF-low population from biological duplicates was analyzed by casTLE[100].

## Single-molecule fluorescence in situ hybridization with immunofluorescence (smFISH-IF)

The detailed protocol for smFISH IF assay was described in detail[101]. Briefly, 20mer single-stranded DNA oligoes were ordered from Integrated DNA Technology in a 96-well plate format. The oligoes were pooled together and labeled with Cy3-NHS ester (Lumiprobe 11020) according to the described protocol[102]. Cells were seeded on 18 mm cover glass and treated with 500 µM indole-3-acetic acid (Sigma, I2886) to degrade AID-labeled reporters on the day before the experiment. During the experiment, the cells were fixed with 4% PFA (Electron Microscopy Sciences, 50-980-492) for 10 min, blocked in 50 mM Glycine (Thermo Fisher Scientific, BP 381-1), in phosphate buffer saline (PBS, Corning, 46-013-CM) buffer for 10 min, then permeabilized with 0.1% Triton-X (Sigma-Aldrich, T8787), and blocked with BSA 5 mg/mL (VWR, VWRV033) for 10 min, all at room temperature. The cells were pre-equilibrated with hybridization buffer (10% formamide, 2×SSC (Roche, 11666681001), BSA 5 mg/mL), and incubated with 100 nM FISH probes targeting transcript of interest and primary antibodies targeting the sf-GFP (Aves Labs, GFP-1010) in hybridization buffer (10% formamide (Acros Organics, AC205821000), 2xSSC, BSA 2 mg/mL) for 3 h at 37 °C. The excess probes and antibodies were washed away with wash buffer (10% formamide, 2×SSC), and the samples were incubated for 20 min at 37 °C with secondary antibodies (Goat anti-Chicken IgY (H+L) Alexa Fluor 647, Thermo Fisher Scientific, A21449). Finally. The samples were washed with PBS and mounted on a microscope slide using ProLong Diamond Antifade Mountant containing DAPI for nucleus staining (Thermo Fisher Scientific, P36962).

## Microscopy

The fixed smFISH-IF samples were imaged on an automated inverted Nikon Ti-2 wide-field microscope equipped with a 60× 1.4NA oil immersion objective lens (Nikon), Spectra X LED light engine (Lumencor), and Orca 4.0 v2 scMOS camera (Hamamatsu).

The live cell imaging was conducted on a customized wide-field microscope built around a Nikon Ti-E stand. The microscope was equipped with three separate EMCCD cameras (Andor iXon Ultra 897) with ultra-flat 2 mm thick imaging splitting dichroic mirrors (T565LPXR-UF2, T640LPXR-UF2). Each of the cameras was equipped with band pass emission filter (ET525/50m, ET595/50m, and ET655lp). The light source of the microscope is an LU-n4 laser launch (Nikon) with four wavelengths: 405, 488, 561, and 640 nm. The excitation was done through H-TIRF with 100× 1.49NA oil immersion TIRF objective (Nikon). The excitation laser and fluorescence emission were separated by a quad band dichroic mirror (Chroma, ET-405/488/561/640 nm). The microscope was also equipped with an

automated linear encoded XY-stage with Piezo-Z (Applied Scientific Instrumentation).

## Live-cell imaging

**Imaging steady state translation.** Cells stably expressing reporters were seeded in 35 mm cover glass (Cellvis, D35-20-1.5-N) and grown overnight. Then, 500 μM 3-Indoleacetic acid (Sigma, I2886) was added to the culture medium throughout the experiment. One hour before imaging, cells were incubated with 100 nM Janelia Fluor Halo dye (JF646, provided by Lavis lab) for 30 min[103] and washed with warm DMEM for 30 min to remove excessive dyes. Prior to imaging, the medium was changed to Leibovitz's L-15 media without phenol red (Gibko, 21083-027) supplemented with 10% FBS. The sample was kept at 37 °C with humidity controlled by the Tokai Hit stage top incubator during the imaging session. For transient transfection of translation reporters, 400 ng plasmids were transfected 12 h before imaging with X-tremeGene HP transfection reagent (Roche, 6366236001). The rest of the protocol is identical to the imaging of stably integrated reporters.

**Ribosome runoff experiment.** Cells with translation sites were identified and imaged. Half of the warm imaging media was mixed with harringtonine (Cayman Chemical, #15361) and added back to the imaging dish to a final concentration of 3 μg/mL. Live imaging was started exactly 1 min after adding the drug. The RNA and protein channel were acquired every 10 s for 30 min.

**Fluorescence recovery after photobleaching for single translation sites.** After identifying the target translation sites, cells were imaged for 30 s prior to bleaching the translation site with a focused 488 nm laser controlled by Bruker microscanner. After beaching the site, the cell was imaged every 5 s for 10 min.

## Data analysis

**smFISH-IF data analysis.** All fixed cell imaging data analysis was performed using existing and custom Matlab programs as described[101]. In brief, the coordinates and amplitudes of smFISH and IF signals were identified independently using FISH-Quant[104]. FISH-Quant uses Gaussian fitting to determine the spot localization and its intensity amplitude. After the identification of single mRNA signals, a bounding box of $11 \times 11$ pixels was drawn around the cytoplasmic mRNA and checked as a potential translation site. The transcription site fitting algorithm of FISH-Quant was used to assess the translation side intensity of the corresponding mRNA. To determine the number of nascent peptides per transcript, the single mature SunTag proteins in the cytoplasm were identified in a similar fashion as RNA transcripts, and the median of single protein intensities was used to normalize the integrated intensities of translation sites.

**Tracking translation of single mRNAs in live cells.** All live cell imaging analysis was performed using existing software packages written in Matlab (MathWorks) supplemented with custom script. Single particles of RNA or translation sites were identified with Airlocalize[105] and tracked by u-track[106]. Only tracks longer than five frames were further analyzed. Translation sites and mRNAs that overlap in space and time are linked together. A temporal overlap of at least five frames was required to link RNA and protein tracks. All tracks were inspected, and mRNAs with merging or splitting events were discarded.

For the ribosome runoff experiment, we measured the time for each translating mRNA to completely lose its translation intensity, starting from the moment of adding harringtonine. This is defined as ribosome runoff time: t (Fig. 4d, e). Practically, if the translation site intensity fell below 10% of the maximum intensity of the trace, it was deemed runoff. If TLS and the corresponding mRNA disappear together, the molecules were not included in the analysis because

these events may be loss of the tracked mRNA. The RNA with low initial protein signal in the first four frames, below 10% of maximum protein intensity, were discarded, as they were not visibly translating at the beginning of the assay. We compiled all runoff times into a translation site survival probability curve: the percentage of translating mRNAs was plotted as a function of time after adding Harringtonine. We used GraphPad Prism software to plot the results of runoff assays as Kaplan–Meier curves at the same time. We used Greenwood's formula to calculate the 95% confidence bounds in Fig. 4. To test the statistical significance between the survival curves, we performed Cox proportional hazards analysis at 15 min using GraphPad Prism (v8.2.1)[107].

**Calculating elongation time through repeats.** The repeats are just a small fraction of the reporter, but the measured ribosome runoff time depends on elongation through the whole reporter. We therefore estimated the elongation time through the GGGGCC repeats, which is inversely proportional to the elongation speed, using the measured ribosome runoff time. Theoretically, the runoff time is the time it takes the last ribosome to translate through the entire construct. We used the median of the runoff time to estimate the total elongation time $t$.

Assume that the lengths of the inserted sequence (repeat or NLuc) and the rest of the construct are $L_I$ and $L_O$, respectively (Fig. 4g). The elongation speed through the insert is $v_I$ and through the rest of the protein is $v_0$. The time spent on the insert and the rest of the protein is $t_O$ and $t_I$, respectively. The total runoff time is equal to the sum of elongation times through the two regions.

$$t = t_I + t_0 = \frac{L_I}{v_I} + \frac{L_0}{v_0} \tag{S1}$$

Rearrange the equation, we can derive

$$\frac{v_0}{v_I} = \frac{t - \frac{L_0}{v_0}}{\frac{L_I}{v_0}} \tag{S2}$$

which is the ratio between elongation speeds of a general protein sequence and of repeats. Note the variables at the right-hand side of the equation are all experimentally measurable quantities. For NLuc, the median runoff time is $t = 4$ min 55 s, from which we can estimate the elongation speed of $v_0 = 3.6 AA/s$, consistent with previously published data[58,59,73]. The elongation speeds through different frames can be estimated according to Eq. S2.

## Proteomic data analysis

The analysis of proteomic data followed the previous pipeline[108]. Briefly, the data matrix was extracted from the Answer ALS database, and the iPSN lines used are listed in Supplementary Data 2. The acquisition of AALS data followed the protocols approved by the Johns Hopkins Institutional Review Board. The protection of health information was in accordance with HIPAA and the HITECH Act. Proteins that had more than four missing values across samples were removed. The protein levels were normalized to the AE8 iPSN batch control line for each batch. *$P < 0.05$, Wilcox test.

## Reporting summary

Further information on research design is available in the Nature Portfolio Reporting Summary linked to this article.

## Data availability

The data that support the findings of this study are available from the corresponding authors upon reasonable request. The source data underlying Figs. 1–6, Supplementary Figs. 1–7 are provided as a Source Data file. The CRISPR screen sequencing data (Fig. 6a) is deposited in

the NCBI BioProject database under the accession number PRJNA905075. Source data are provided with this paper.

## Code availability

The custom code used in the analysis is freely available at https://github.com/binwulab/.

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

## Acknowledgements

We thank the Sun and Wu lab members for their helpful discussion. This work is supported by Johns Hopkins University School of Medicine Discovery Fund Synergy Award, National Institute of Health (RF1NS113820 to S.S. and B.W.; R01GM136897 to B.W.; R01NS107347, R21AG072078 and RF1NS127925 to S.S.), National Science Foundation (MCB 1817447 to B.W.), Pew Charitable Trust (00030601 to B.W.), and the Packard Center for ALS Research. M.J.L. was supported by the NIH Training Grant (T32 GM008403), and N.M.L. and B.L. were supported by the NIH Training Grant (T32 GM007445).

## Author contributions

M.J.L., S.W., D.D., S.S. and B.W. contributed to the experimental design and interpretation of the study and wrote the manuscript with input from other authors. M.J.L., S.W., D.D., B.N., N.M.L. and R.W. conducted all experiments under the mentorship of S.S. and B.W. N.Z., T.J.S. and M.C.B. edited the manuscript and provided key reagents.

## Competing interests

The authors declare no competing interests.

## Additional information

[1]Department of Biophysics and Biophysical Chemistry, Johns Hopkins University School of Medicine, Baltimore, MD 21205, USA. [2]Center for Cell Dynamics, Johns Hopkins University School of Medicine, Baltimore, MD 21205, USA. [3]Department of Physiology, Johns Hopkins University School of Medicine, Baltimore, MD 21205, USA. [4]Brain Science Institute, Johns Hopkins University School of Medicine, Baltimore, MD 21205, USA. [5]Department of Biochemistry and Molecular Genetics, University of Colorado-Anschutz Medical Campus, Aurora, CO 80045, USA. [6]Department of Genetics, Stanford University School of Medicine, Stanford, CA 94305, USA. [7]Solomon H. Snyder Department of Neuroscience, Johns Hopkins University School of Medicine, Baltimore, MD 21205, USA. [8]Department of Pathology, Johns Hopkins University School of Medicine, Baltimore, MD 21205, USA. [9]These authors contributed equally: Malgorzata J. Latallo, Shaopeng Wang, Daoyuan Dong. ✉e-mail: shuying.sun@jhmi.edu; bwu20@jhmi.edu

