## [Peer Review File · Nature Communications]

Single molecule imaging reveals distinct elongation and frameshifting dynamics between frames of expanded RNA repeats in C9ORF72-ALS/FTDREVIEWER COMMENTS

Reviewer #1 (Remarks to the Author):

The study by Latallo et al. demonstrates a serious endeavor to explore the pathogenic mechanism of repeat expansion, namely the accumulation of DPR proteins. An impressive number of constructs were generated to obtain high quality single molecule images that report on translation dynamics in vivo, both for translation initiation and elongation. The authors not only convincingly show the change in the translation rate between the 3 different frames investigated, but also show the change in frameshift rate using imaging techniques which produced compelling 2 color images. Of particular interest was the question of whether the alternate frames occurred as a result of the GA frameshift, which the authors demonstrate is not the case, and rather occur from independent events. Interestingly, examination of the specific amino sequences present, showed how the presence of unique amino acids affects the translation rates, rather than the secondary structure which is often a primary component. Lastly, the authors identified specific proteins in the RQC pathway that modulate the translation dynamics, and the ways in which ribosome collision affects protein abundance, and how this interplays with canonical or non-canonical initiation. Altogether, this study is important and convincing, and would be suitable for Nature Communications, pending the following comments.

General remarks:

* The writing is somewhat jumpy and at times repetitive, and could be improved. Grammar must be checked since in some parts there are significant text problems (e.g. pages 8-9).

* Authors should consider moving some of supplemental figures to be main figures, as they are not “supplemental” to the story. For instance, Figure S1; Pelota data in Figure S7, which is shown to have a role here in tandem with the ZNF598, and the role of RAN translation shown in Figure S4. Other parts of the data could be considered as well.

Major issues:

* Abstract: This sentence is unclear – “Besides initiation, translation elongation rates vary drastically between different frames, with GP slower than GA, and GR the slowest.” – it is not clear at this point of the abstract what are GP/GA/GR, and which frames/frameshifts are you referring to.

* The system is quite complex and I encourage the authors to re-read this first section that introduces the system to see how they can further clarify. For instance, it is not well explained how one gets the 3 different frames in the first place. In connection to that, Fig. S1c shows the percentage of translating RNAs that potentially translate either GA, or GP or GR, but there is no actual proof that indeed these specific di-peptide sequences are being produced.

* Figure S1:

1) Why 70 repeats? The authors state that in healthy people there are less than 10 repeats, while it can reach thousands in sick patients. Did the authors check to see if there is a difference in frameshifting of for example 10 repeats vs 100 or even 1000 repeats?

2) Page 6: line 2- it is not clear if the FISH probes hybridize with the MS2 loops or with a GFP sequence, please clarify.

3) Line 6: why would the fully synthesized proteins be dimmer? Maybe it is just the scFv-sfGFP background? Perhaps add a comparison to the appearance of a cell expressing just scFv-sfGFP as a

control.

4) Figure S1c: Is each shape indicative of a different biological replicate? This should be indicated.

5) The authors say that the “The fraction of RAN translating mRNA was significantly lower than regular mRNAs (~80%) but higher than the negative control (without insertion of repeats).”, but the data for the negative control is not presented. Is this a novel finding or a previous finding?

6) S1f: is there data for GP? Because GP showed higher translation in the experiment performed in S1h, it would be of interest to see how it looked compared to the GA here. Is there a statistically significant change between GA and GR? Data is only shown for the GA and GR mutants, which comes close to significance but not exactly there.

7) S1h: authors cite ‘significant’ change, if this refers to statistical significance, perhaps it should be presented.

8) One of the aims was to determine the translation efficiency of the GA frame, and if this accounted for the variability in DPR abundance. Overall figure S1 presents very interesting constructs and presents the translation data with multiple methods to claim that the GA reporters show increased translation efficiency when compared to the GP and GR. However, the presence of ‘one star’ significance in figure S1c highlights their absence in the others, as well as the missing GP data in S1f. It appears the data is robust and reliable, but it does cause a pause for thought. This is amplified when we learn later in the story that the “difference in the initiation rate was insufficient to explain the measured difference in the final DPR amounts”, and that it is really the elongation rates that dictate some of the other features discussed, and in general the presence of the other frames probably come from independent events, rather than GA efficiency.

* The basis of the live-cell imaging of mRNP translation is the tethering of the MS2-CP to cell membrane. Could the localization of translation by the membrane and not in the cytosol have any effect on the conclusions of the study? Regarding this point, it is unclear if in the Harringtonine experiments in Fig. 3 is the mRNA also tethered?

* Fig. 1 (page 9) - How was the time of frameshift measured? Is it a time from the appearance of the green signal? or the amount of time there is the Suntag signal in the movie? Could the authors show the difference in time between the appearance of a full HA signal (top magenta peek) to the appearance of a full Suntag signal (top green peek), and compare GA,GP,GR. Perhaps it takes more time to shift from GA to GR then from GR to GA, and that could strengthen their point.

* Fig. 2e-f – in addition to quantifying the percentage of frameshift events the authors could extract additional information such as the average time of translating in the other frame, the likeliness that a frameshift would occur more than once on a specific RNA, does the frameshift reverse back to the regular frame after the frameshift phase (do you get a purple line after a green line?)

* Figure 5c: It is unclear what the fluorescent micrographs are meant to display (and the figure should indicate what each color signifies). While the survival graphs demonstrate control vs siRNA for ZNF598, it is hard to discern the exact takeaway from the images in regard to ribosomal runoff-times.

* Repeat expansion might cause RNA toxicity via the sequestration of RNA-binding proteins (PMID 23381195) which could affect the translation. Could the authors speculate about this with regard to their systems.

Minor:

- * Page 3 – Introduction – “are more widely observed in patient post-mortem tissues” – widely?
- * Page 3 – Add some information about RAN translation (phosphorylation of eIF2A, cap+eIF4A dependent...) (e.g. PMID 29222490).
- * Page 5 - end of Introduction – “The insight obtained from this single molecule study is instrumental in understanding the disease etiology of C9ORF72-ALS/FTD.” – how this is instrumental is unclear.
- * Page 5 – Results – 2nd line - "(Fig. S1a): 70x (GGGGCC)" – in the figure it is called WT.
- * Page 5 -Results – “poly-GA” – poly
- * End of page 6: GR in S1f looks more close to 20-15% than to 10%. And how do the authors interpret $P=0.0535$?
- * Page 7 line – Fig. S1f (not 1f)
- * Page 7 – “Surprisingly, the difference between frames, especially between GA and GR, in ensemble measurements is higher than for single molecule measurements” – unclear, which differences, which ensemble measurements are you referring to? Also, it is unclear how this conclusion was reached.
- * Page 7 line 5- significantly higher – where is the statistical analysis.
- * Fig. S1a – bottom scheme, one arrow was swept away...
- * Page 9 - Beginning of the 2nd paragraph – would be helpful to have some kind of scheme/table like in S2a.
- * Fig. 1a – maybe draw a line to show where DPR1/2 start and end.
- * Consider adding controls to top of S2a.
- * Legends of S2d and S4d are skew (in the figures).
- * Figure S3: Trace plots are interesting visually but do not offer much information to the reader. These should be quantified as presented in Fig 1g.
- * Figure 3c: What is the diffuse background signal seen in the GR significant but not in the other 2 frames.
- * Page 12 - line 7 from the end- why is Proline mentioned? Please add an explanation including for Arginine.
- * Page 13 - AUG-HiBiT – explain
- * Page 15 - what does "(GGGGCC)n-encoded" stand for?
- * Live-cell movies would be enhanced if colored labels were added indicating what each signal represents.
- * The legend of the movies mentions they are 30 min long but the actual movies shown are shorter. The legend should mention the actual times presented.

Reviewer #2 (Remarks to the Author):

Latallo, Wang, Dong et. al. examined factors that contribute to dipeptide repeat protein production in

C9ORF72 amyotrophic lateral sclerosis (C9ALS/FTD). The authors established polyclonal immortal cell lines (HeLa, U2OS and SH-SY5Y) stably expressing exogenous GGGGCC-DPR reporters to examine translation elongation rate of dipeptide repeat proteins (DPRs) using single molecule imaging technologies for GA-DPR being translated the fastest, followed by GP-DPR and GR-DPR. Using the same system, the authors detected repeat frameshifting occurred, but at a relatively low frequency. CRISPR-Cas9 knockout screening was also performed in an attempt to find out factors that modulate DPR production, and ZNF598 and Pelota were screened out as two prominent DPR production modulating factors. These are all novel findings and will help advance the field of repeat expansion-mediated RAN translation in general. Some of the experimental results are not conclusive and will need to be further validated. The reviewer also raises concerns about how these new data can enlighten the C9ALS/FTD field in particular, especially DPR-mediated toxicity in the disease.

Major concerns:

- 1) The disease relevance of this work needs to be further enhanced. It is necessary for the authors to demonstrate their findings in monoclonal iPSC-derived neurons stably expressing respective DPR reporters. It would be ideal if their findings can further be demonstrated in C9ALS/FTD patient iPSC-derived neurons.
- 2) Do the AID degron 24, SunTags and MS2-binding sites sequences contain any repeat sequences? If so, would these non-GGGGCC repeat sequences encoding these tags alter RAN translation of reporters making them behaving differently? How would the authors ensure what were detected in the reporter are reflecting the DPR production of the endogenous intronic GGGGCC sequences?
- 3) Did the authors conduct the experiments using monoclonal stable HeLa, U2OS and SH-SY5Y cell lines? If only polyclonal stable lines were produced, the expression level of the transcripts may vary between individual cells in the cell population due to the presence of potentially multiple insertion sites of the reporter constructs in different stably-transfected cells in the cell population. How would the authors control such variance?

In some figures, the variance of biological replicates appears to be huge. For example:

Fig. S1C: The deviation between biological replicas appears to be large. Will this be because of the polyclonal nature of the stable cell lines? Also, without normalization between the different individual RAN reporter cell lines (GA, GP and GR) and other controls, it is difficult for readers to determine whether the difference is due to the intrinsic properties of RAN, variation due to transgene integration.

The data were mean \pm standard deviation from three biological replicates for wild type cell lines (GA: 82, 67, 29 cells; GP: 51, 33, 26 cells; GR: 61, 53, 25 cells).

In Fig. S1f, the three biological replicates vary hugely. This could reflect the heterogeneity of the cells.

Each symbol represented a single RNA and the three shapes represented three biological replicates (GA: 43, 105 and 150 translating RNA molecules; GA-CGG: 81, 14, 91 translating RNAs; GR: 103, 98, 99 translating RNAs, GR-CGG: 170, 33, 119 translating RNAs).

4) The authors must demonstrate the role of translation elongation in the pathogenic mechanisms of the C9ALS/FTD disease. For instance, other than their involvement in DPR production, do ZNF598 and Pelota also play some roles in mediating C9ALS/FTD toxicity?

Minor comments:

The arrangement of some of the figures does not correspond to the text. For example, Fig. 1f appears before Fig. 1a in the text. Such arrangements increased the difficulty of reading the manuscript.

Reviewer #3 (Remarks to the Author):

In this work, several insights are made regarding DPR synthesis from G4C2 repeats. Sophisticated techniques, including single molecule translation imaging, are used appropriately and rigorously to assess facets of DPR initiation and elongation. More traditional, “bulk” luciferase-based methods are used to confirm their findings. The CRISPR screen of 1906 proteins identified RQC factors involved in DPR translation and the findings supported through functional assays. Overall, the methodologies are elegant, the experimental design is rigorous, and interpretations of data are well-reasoned. This is a nice, scientifically sound mechanistic study.

With that said, DPRs have been studied extensively and the current findings are incremental. The findings add little new information and mostly confirm what has already been observed. They are thus of nominal significance to the field, in the reviewer’s opinion. For example, the observation that frame-shifting can occur had already been made, reducing novelty of the current study. The identification of ZNF598 and Pelota as modulators of DPR translation is interesting, but there is little evidence or discussion of how the mechanism could potentially be targeted as a therapeutic approach. It is frustrating that the contribution of DPRs to disease remains controversial and that the current study does not reduce ambiguity.

Response to reviewers

We would like to thank all the reviewers for the constructive comments and suggestions that help us improve this manuscript. We included additional analysis of existing data and enriched the manuscript with more experiments and controls. The major modifications were marked with yellow highlights in the Results, Method, and Figure legends.

Reviewer #1 (Remarks to the Author):

The study by Latallo et al. demonstrates a serious endeavor to explore the pathogenic mechanism of repeat expansion, namely the accumulation of DPR proteins. An impressive number of constructs were generated to obtain high quality single molecule images that report on translation dynamics in vivo, both for translation initiation and elongation. The authors not only convincingly show the change in the translation rate between the 3 different frames investigated, but also show the change in frameshift rate using imaging techniques which produced compelling 2 color images. Of particular interest was the question of whether the alternate frames occurred as a result of the GA frameshift, which the authors demonstrate is not the case, and rather occur from independent events. Interestingly, examination of the specific amino sequences present, showed how the presence of unique amino acids affects the translation rates, rather than the secondary structure which is often a primary component. Lastly, the authors identified specific proteins in the RQC pathway that modulate the translation dynamics, and the ways in which ribosome collision affects protein abundance, and how this interplays with canonical or non-canonical initiation. Altogether, this study is important and convincing, and would be suitable for Nature Communications, pending the following comments.

General remarks:

* The writing is somewhat jumpy and at times repetitive, and could be improved. Grammar must be checked since in some parts there are significant text problems (e.g. pages 8-9).

Response: We appreciate the reviewer for the positive comments about the quality of the study. We worked hard in the revision to improve the clarity. We have marked major changes with yellow highlights. But there are also many small unmarked edits to smooth the text. We hope it will increase the readability of the manuscript.

* Authors should consider moving some of supplemental figures to be main figures, as they are not “supplemental” to the story. For instance, Figure S1; Pelota data in Figure S7, which is shown to have a role here in tandem with the ZNF598, and the role of RAN translation shown in Figure S4. Other parts of the data could be considered as well.

Response: We thank the reviewer for this important suggestion. In the original submission, we were not sure whether Figure S1 diluted out the major conclusion and S7 was too repetitive. Encouraged by the reviewer’s suggestion, we have rearranged our manuscript significantly. We moved the majority of the translation initiation results (Original Figure S1) to the current Figure 1. We also moved RAN translation results from Figure S4 to current Figure 3 and Pelota data to the current Figure 6.

Major issues:

* Abstract: This sentence is unclear – “Besides initiation, translation elongation rates vary drastically between different frames, with GP slower than GA, and GR the slowest.” – it is not clear at this point of the abstract what are GP/GA/GR, and which frames/frameshifts are you referring to.

Response: We introduced the concept of multiple polypeptides being produced early on in the abstract to improve clarity.

* The system is quite complex and I encourage the authors to re-read this first section that introduces the system to see how they can further clarify. For instance, it is not well explained how one gets the 3 different frames in the first place. In connection to that, Fig. S1c shows the percentage of translating RNAs that potentially translate either GA, or GP or GR, but there is no actual proof that indeed these specific di-peptide sequences are being produced.

Response: The reviewer has brought up an important conceptual assumption that we and other investigators in the field have made implicitly. We placed reporters after repeats to report the in-frame dipeptide production. We would like to point out that each reporter measures one RAN frame only. We built different reporters to visualize different reading frames separately. In addition, it does not disqualify translation in other frames, which can happen at the same time but are not visualized. We are also aware that this is not the direct proof that the specific di-peptide are being produced because frame-shift can occur. Nevertheless, we have shown later that the frameshift is rare. So on the first-order approximation, we assumed that the reporter represents the RAN translation. With this limitation in mind, we edited the text according to reviewers' suggestions (marked in yellow) and discussed this point in the Discussion section.

* Figure S1:

1) Why 70 repeats? The authors state that in healthy people there are less than 10 repeats, while it can reach thousands in sick patients. Did the authors check to see if there is a difference in frameshifting of for example 10 repeats vs 100 or even 1000 repeats?

Response: We thank the reviewer for raising this issue, critical for interpreting our experimental results. We used 70 repeats as this is the maximum length that can be stably cloned in the reporter constructs. It is technically challenging to create repeat stretches beyond that as long repeats are less stable and randomly truncated. It's an interesting question to check the difference in frameshifting of different repeat lengths. In principle, the longer the repeats, the higher chances to have frameshift. But it is technically difficult to verify this. We realized this point and discussed it in the Discussion session.

2) Page 6: line 2- it is not clear if the FISH probes hybridize with the MS2 loops or with a GFP sequence, please clarify.

Response: The FISH probes hybridize with the MS2 binding sites and SunTag region on the reporter RNAs. scFv-sfGFP is expressed by separate mRNAs and we were not probing it. We use the scFv-sfGFP protein only to label translation sites. We edited the text according to reviewers' suggestions, marked in yellow.

3) Line 6: why would the fully synthesized proteins be dimmer? Maybe it is just the scFv-sfGFP background? Perhaps add a comparison to the appearance of a cell expressing just scFv-sfGFP as a control.

Response: We thank the reviewer for raising this point. A translating mRNA may have multiple ribosomes, therefore multiple nascent peptides on it at the same time, while a mature protein only carries a single SunTag. Therefore, the fully synthesized proteins are dimmer, and translation sites are brighter. A single

SunTag carries 24 copies of epitope and can bind up to 24 scFv-sfGFPs. Therefore, a single mature protein is still visible while a single scFv-GFP would be too dim to be seen. To show the difference between the background scFv-sfGFP and mature single proteins, we added a supplemental figure showing the IF image of cells containing scFv-sfGFP, but no reporter (Fig. S1b). As one can see, there is no mature SunTag spot at all. We have edited the text to make this point more clear.

4) Figure S1c: Is each shape indicative of a different biological replicate? This should be indicated.

Response: We thank the reviewer for pointing this out. We edited the text according to reviewers' suggestions, marked in yellow in the legend of Figure 1c.

5) The authors say that "The fraction of RAN translating mRNA was significantly lower than regular mRNAs (~80%) but higher than the negative control (without insertion of repeats).", but the data for the negative control is not presented. Is this a novel finding or a previous finding?

Response: We agreed with the reviewer that this is an important control experiment. To clearly demonstrate the dynamic range of the measurement, we now added both positive and negative controls (Fig 1 c). The translation of regular mRNA has been published previously (Positive Ctrl, Wu et al, 2016 Science), now reproduced here. The negative control without repeat insert is a new experiment data (Fig. 1c, S1a, c). There is negligible translation without repeats. We edited the text and adjusted the figure legend to accommodate those changes.

6) S1f: is there data for GP? Because GP showed higher translation in the experiment performed in S1h,@ it would be of interest to see how it looked compared to the GA here. Is there a statistically significant change between GA and GR? Data is only shown for the GA and GR mutants, which comes close to significance but not exactly there.

Response: We thank the reviewer for asking for clarification. At the time of submission of this manuscript we were unable to provide the data for GP cell line. We now have generated the GP reporter constructs suitable for the live cell imaging experiment and added the data from the translation of GP and GP-CGG constructs. We also adjusted the text and figures accordingly (Fig. 1f). When CUG was mutated to CGG, the fraction of translation time for GP frame did not change significantly, similar to GR. This supports that GP and GR frames were not due to frameshifted from ribosomes initiated from CUG near-cognate start codon.

7) S1h: authors cite 'significant' change, if this refers to statistical significance, perhaps it should be presented.

Response: We thank the reviewer for this important point. We have performed a statistical significance test and presented it in Figure S1e.

8) One of the aims was to determine the translation efficiency of the GA frame, and if this accounted for the variability in DPR abundance. Overall figure S1 presents very interesting constructs and presents the

translation data with multiple methods to claim that the GA reporters show increased translation efficiency when compared to the GP and GR. However, the presence of ‘one star’ significance in figure S1c highlights their absence in the others, as well as the missing GP data in S1f. It appears the data is robust and reliable, but it does cause a pause for thought. This is amplified when we learn later in the story that the “difference in the initiation rate was insufficient to explain the measured difference in the final DPR amounts”, and that it is really the elongation rates that dictate some of the other features discussed, and in general the presence of the other frames probably come from independent events, rather than GA efficiency.

Response: We appreciate the reviewer for acknowledging the intricacy of DPR production. We would like to emphasize the previously neglected translation elongation. That was why we placed all the initiation data in the supplemental figure in the original submission. Now, we followed the reviewer’s suggestion and moved the data in Fig S1 to main Figure 1. We thank the reviewer for the proposition to go deeper to compare the RAN initiation efficiency. According to the reviewer’s suggestion, we provided additional data in Figure 1f showing the results for GP and the GP-CGG mutant. Like GR, the CGG mutation did not reduce GP translation (Fig. 1f), suggesting that both GP and GR DPRs did not result from the upstream CUG initiation in the GA frame.

* The basis of the live-cell imaging of mRNP translation is the tethering of the MS2-CP to cell membrane. Could the localization of translation by the membrane and not in the cytosol have any effect on the conclusions of the study? Regarding this point, it is unclear if in the Harringtonine experiments in Fig. 3 is the mRNA also tethered?

Response: We thank the reviewer for asking for clarification. Yes, mRNAs are tethered to the plasma membrane in all live cell experiments, including the Harringtonine Run Off. The purpose of tethering is to allow tracking of single mRNAs for a long time (30 minutes in our experiments). The MS2-MCP tethering has been used in a number of publications previously (Yan et al, Cell 2016, Goldman et al, Mol Cell, 2021, Livingston et al, BioRxiv, 2023). It has been shown that it did not change ribosome loading significantly. Moreover, the only change between different reporters is the insertion of one or two nucleotides to report different reading frames. The rest of mRNA, including the 3’UTR and MS2 tethering, is the same. So we believe the relative change between different reporters is due to different DPR, not due to tethering. We edited the manuscript to explicitly state the usage of tethered system in Run Off as well as in live cell tracking experiments.

* Fig. 1 (page 9) - How was the time of frameshift measured? Is it a time from the appearance of the green signal? or the amount of time there is the SunTag signal in the movie? Could the authors show the difference in time between the appearance of a full HA signal (top magenta peek) to the appearance of a full SunTag signal (top green peek), and compare GA, GP, GR. Perhaps it takes more time to shift from GA to GR then from GR to GA, and that could strengthen their point

Response: We thank the reviewer for asking for clarification. The frameshift events were defined as the appearance of the green signal in the presence of magenta. In the revised manuscript, we also quantified the fraction of time that each translation events occupy (Figure S3d). We thank the reviewer for suggesting additional quantification of our data. Unfortunately, the signal in HA and SunTag may come from multiple

ribosomes translating at the same time. Occasionally, one of ribosomes translating HA-tag may shift to SunTag frame. It is hard to correlate the maximal intensity of HA peak to SunTag.

* Fig. 2e-f – in addition to quantifying the percentage of frameshift events the authors could extract additional information such as the average time of translating in the other frame, the likeliness that a frameshift would occur more than once on a specific RNA, does the frameshift reverse back to the regular frame after the frameshift phase (do you get a purple line after a green line?)

Response: We thank the reviewer for the suggestion. We performed additional analysis of translation time as suggested by the reviewer (Fig. S3d). However, in most cases, we do not have a single ribosome translating an mRNA. The HA translation is driven by AUG start codon with high initiation rate. Many ribosomes translating in HA frame increase the chances of observing the rare frameshift events. But at the same time it also makes it impossible to distinguish which ribosome shifted to another frame. The current platform does not allow us to observe the frameshift-reversing-back event, because once the ribosome shifted back to HA frame from SunTag frame, we would not be able to detect it. Instead, by constructing different reporters to measure different frameshift scenarios, we hope to measure the frameshifting probabilities between different frames at the first order approximation.

* Figure 5c: It is unclear what the fluorescent micrographs are meant to display (and the figure should indicate what each color signifies). While the survival graphs demonstrate control vs siRNA for ZNF598, it is hard to discern the exact takeaway from the images in regard to ribosomal runoff-times.

Response: We thank the reviewer for helping to improve the annotation. We wanted to show that the runoff is even slower under ZNF598 KD. The difference is subtle because the runoff in GR frame is already slow at baseline. We now included images with SunTag signal only for more clear visualization and edited the figure legends with better annotation, hopefully delivering the message (Fig. 6c).

* Repeat expansion might cause RNA toxicity via the sequestration of RNA-binding proteins (PMID 23381195) which could affect the translation. Could the authors speculate about this with regard to their systems.

Response: Indeed, it is challenging to separate RNA toxicity when we consider translation and dipeptide. As the reviewer suggested, repetitive RNA may sequester RNA-binding proteins (RBP). These RBPs may regulate RNA splicing, export, stability as well as translation. It is not surprising that RBPs might influence the translation initiation, elongation, termination or frameshift, possibly by influencing the repeat RNA structures. We added to the discussion.

Minor:

* Page 3 – Introduction – “are more widely observed in patient post-mortem tissues” – widely?

Response: Thank you, we now corrected it!

* Page 3 – Add some information about RAN translation (phosphorylation of eIF2A, cap+eIF4A dependent...) (e.g. PMID 29222490).

Response: We have added references and discussed the influences of translation initiation factors and integrated stress response on RAN translation in the introduction.

* Page 5 - end of Introduction – “The insight obtained from this single molecule study is instrumental in understanding the disease etiology of C9ORF72-ALS/FTD.” – how this is instrumental is unclear.

Response: We changed the sentence to: “The insights gained from this single molecule study are instrumental in understanding the biogenesis of DPR and their contribution to the disease. Modulation of translation is a promising therapeutic approach to prevent accumulation of toxic DPRs”.

* Page 5 – Results – 2nd line - "(Fig. S1a): 70x (GGGGCC)" – in the figure it is called WT.

Response: Thank you! We now changed it to “wild type 70x (GGGGCC)”.

* Page 5 -Results – “poy-GA” – poly

Response: Thank you! We have now corrected it!

* End of page 6: GR in S1f looks more close to 20-15% than to 10%. And how do the authors interpret P=0.0535?

Response: We thank the reviewer for pointing this out. Now we added additional data for the GA and GA CGG sets. As another reviewer pointed out that there was a large variation among previously collected data from the GA reporter cells. We ran an outlier test on all four GA biological replicates (previously collected three and one new). One of the old biological replicate sets was identified as an outlier. We decided to remove that set and the corresponding GA CGG set, as that replicate is not representative due to the low number of data points. The current set comparison shows ** significance (P = 0.0071). We also added additional GP and GP CGG data sets, which did not show a significant difference between each other, similar to GR and GR CGG. All text associated with that figure was edited accordingly.

* Page 7 line – Fig. S1f (not 1f)

Response: Thank you, we have now corrected it!

* Page 7 – “Surprisingly, the difference between frames, especially between GA and GR, in ensemble measurements is higher than for single molecule measurements” – unclear, which differences, which ensemble measurements are you referring to? Also, it is unclear how this conclusion was reached.

Response: Thank you for bringing up this point. The ensemble measurement is referred to the luciferase assay in Fig. 1e. We meant the different DPR levels measured by the luciferase approach (GA is >30 fold of GP and >100 fold of GR) is higher than differences measured by the single molecule method. We edited the writing to clarify.

* Page 7 line 5- significantly higher – where is the statistical analysis.

Response: Thank you! We have now added the statistics to our figure.

* Fig. S1a – bottom scheme, one arrow was swept away...

Response: Thank you! We have corrected the figure which will now be a main panel.

* Page 9 - Beginning of the 2nd paragraph – would be helpful to have some kind of scheme/table like in S2a.

Response: The frameshift reporter scheme was shown in S2a.

* Fig. 1a – maybe draw a line to show where DPR1/2 start and end.

Response: Thank you for this point. However, the frameshift will happen randomly on the repeat, thus there is no particular start site we could indicate on our reporter.

* Consider adding controls to top of S2a.

Response: Thank you, we have now changed the order of the figure!

* Legends of S2d and S4d are skew (in the figures).

Response: Thank you for pointing this out. We have now corrected it.

* Figure S3: Trace plots are interesting visually but do not offer much information to the reader. These should be quantified as presented in Fig 1g.

Response: Thank you. The percentages of translating mRNAs undergoing normal translation, frameshift, or RAN translation are quantified in Fig. 2g. The fractions of time for each type of translation event are quantified in Fig. S3d. Both information are calculated from traces visualized in S3 a-c. Traces visible in Fig. 2f are 15 min snapshots of two constructs from S3.

* Figure 3c: What is the diffuse background signal seen in the GR significant but not in the other 2 frames.

Response: We thank the reviewer for pointing out the difference between the images. The visible background is the signal from MS2 coat protein tethered to the membrane (red), which has cell-to-cell variation. There will also be a diffusive background signal in the GFP channel, from scFv-sfGFP not bound to translation sites. The variation in background is due to the variable intrinsic expression levels of MCP and scFV of the parental cell line, which we used to put in the different DPR reporters. Therefore, it is

independent from the DPR reporter expressed in the cell. All three movies were processed using the same level of background subtraction in each channel.

* Page 12 - line 7 from the end- why is Proline mentioned? Please add an explanation including for Arginine.

Response: Thank you for pointing that out. We mentioned in the manuscript the influence of the Proline and Arginine to the slower elongation speed, as we observed GP and GR have slower elongation.

* Page 13 - AUG-HiBiT – explain

Response: Thank you. We now explained it.

* Page 15 - what does "(GGGGCC)_n-encoded" stand for?

Response: Thank you. We now corrected it to (GGGGCC)₇₀ encoded GR.

* Live-cell movies would be enhanced if colored labels were added indicating what each signal represents.

Response: Thank you for the suggestion. The color labels were originally in the Movie Legends. We agreed with the reviewer that it would be easier to understand if the movie is directly labeled. Now, we have put labels in the movies whenever possible.

* The legend of the movies mentions they are 30 min long but the actual movies shown are shorter. The legend should mention the actual times presented.

Response: Thank you for the suggestions. We now labeled the length of movies accurately.

Reviewer #2 (Remarks to the Author):

Latallo, Wang, Dong et. al. examined factors that contribute to dipeptide repeat protein production in C9ORF72 amyotrophic lateral sclerosis (C9ALS/FTD). The authors established polyclonal immortal cell lines (HeLa, U2OS and SH-SY5Y) stably expressing exogenous GGGGCC-DPR reporters to examine translation elongation rate of dipeptide repeat proteins (DPRs) using single molecule imaging technologies for GA-DPR being translated the fastest, followed by GP-DPR and GR-DPR. Using the same system, the authors detected repeat frameshifting occurred, but at a relatively low frequency. CRISPR-Cas9 knockout screening was also performed in an attempt to find out factors that modulate DPR production, and ZNF598 and Pelota were screened out as two prominent DPR production modulating factors. These are all novel findings and will help advance the field of repeat expansion-mediated RAN translation in general. Some of the experimental results are not conclusive and will need to be further validated. The reviewer also raises concerns about how these new data can enlighten the C9ALS/FTD field in particular, especially DPR-mediated toxicity in the disease.

Major concerns:

1) The disease relevance of this work needs to be further enhanced. It is necessary for the authors to demonstrate their findings in monoclonal iPSC-derived neurons stably expressing respective DPR

reporters. It would be ideal if their findings can further be demonstrated in C9ALS/FTD patient iPSC-derived neurons.

Response: We appreciate the reviewer's suggestion. DPR toxicity in C9ALS/FTD has been well documented in the literature. However, how DPRs are produced, especially the translation properties on the repeat RNA, is still not completely elucidated. Understanding the biogenesis of DPR is important to understand the disease mechanism of C9ALS/FTD. We chose U-2 OS cell line because these cells are well spread out and suitable for single molecule imaging. We also used luciferase assay to measure translation products in multiple cell types, which gave similar results, suggesting the basic molecular pathways are conserved among cell types. Therefore, we think it is not unreasonable to use model cell lines which are more suitable for single molecule imaging studies. This allows direct measurement of the translation dynamics and properties instead of steady state of final protein products, which can reveal the basic molecular mechanism of repeat RNA translation that cannot be answered by traditional assays. We agree with the reviewer that a disease-relevant cell type would be even better, such as neurons. However, it is technically challenging to perform the single molecule imaging in the iPS-neuron culture due to high background signals, complex morphology and difficulty in integrating auxiliary transgenes. We hope to generalize the research in future studies after the technique is further improved. Nevertheless, we believe our findings provide novel and important insight on the basic mechanisms of repeat RNA translation that is relevant to the disease.

2) Do the AID degron 24, SunTags and MS2-binding sites sequences contain any repeat sequences? If so, would these non-GGGGCC repeat sequences encoding these tags alter RAN translation of reporters making them behaving differently? How would the authors ensure what were detected in the reporter are reflecting the DPR production of the endogenous intronic GGGGCC sequences?

Response: We thank the reviewer for raising this concern. AID degron is a simple non-repetitive protein motif. 24xSunTag and 24xMBS are not 24x repetitive sequences. For SunTag, we have kept the amino acid sequences by using synonymous codons to avoid repetition. For 24xMBSV5, we have used MS2 consensus sequences to generate 24x non-repeating MBS and random linker in between. Moreover, the reporters have the same sequences except an insertion of one or two nucleotide to offset the frame. We carefully designed control plasmids and we focus on the relative differences between different reading frames. Our goal is to study the relative properties of translation of different frames on the repeat RNA and how it is influenced by other modifier proteins. We think the design of the reporters can provide the evidence supporting the findings.

3) Did the authors conduct the experiments using monoclonal stable HeLa, U2OS and SH-SY5Y cell lines? If only polyclonal stable lines were produced, the expression level of the transcripts may vary between individual cells in the cell population due to the presence of potentially multiple insertion sites of the reporter constructs in different stably-transfected cells in the cell population. How would the authors control such variance?

Response: We thank the reviewer for raising this point. We used polyclonal cell lines and quantified large number of cells in order to capture the variation and have meaningful statistic comparison. We believe this is a better strategy than monoclonal cell lines, which often exhibit unusual heterogeneity between clones even from one reporter.

In some figures, the variance of biological replicates appears to be huge. For example:

Fig. S1C: The deviation between biological replicas appears to be large. Will this be because of the polyclonal nature of the stable cell lines? Also, without normalization between the different individual RAN reporter cell lines (GA, GP and GR) and other controls, it is difficult for readers to determine whether the difference is due to the intrinsic properties of RAN, variation due to transgene integration. The data were mean \pm standard deviation from three biological replicates for wild type cell lines (GA: 82, 67, 29 cells; GP: 51, 33, 26 cells; GR: 61, 53, 25 cells).

Response: We thank the reviewer for raising his concerns. We have added positive and negative controls and presented it as the new Fig 1c. The data showed that there are some intrinsic variations in the measurement within the normal technical variation range. This is FISH-IF technical variation, not due to the polyclonal nature of the cell lines.

In Fig. S1f, the three biological replicates vary hugely. This could reflect the heterogeneity of the cells.

Response: We thank the reviewer for pointing out the variation in the presented figure (now Fig. 1f). Reviewer 1 also raised a similar issue: there was a large variation between previously collected GA data from three biological replicates. Now, we added additional data to our GA and GA CGG sets. We ran an outlier test on all four biological replicate sets (previously collected three and one new). One of the old biological replicate sets was flagged as an outlier. We decided to remove that set and the corresponding GA CGG set, as that replicate is not representative due to the low number of data points. The current set comparison shows ** significance ($P = 0.0071$). We also added additional GP and GP CGG data sets requested by the reviewer, which did not show a significant difference between each other, similar to GR and GR CGG. All text associated with that figure was edited accordingly.

4) The authors must demonstrate the role of translation elongation in the pathogenic mechanisms of the C9ALS/FTD disease. For instance, other than their involvement in DPR production, do ZNF598 and Pelota also play some roles in mediating C9ALS/FTD toxicity?

Response: We appreciate the reviewer's suggestion. The goal of the paper is to explain the basic molecular mechanism of repeat RNA translation. The different properties of different reading frames will influence the level of DPRs, which are important contributors for the pathogenesis of C9ALS/FTD themselves. However, we kindly disagree that we must also include evidence of the toxicity mediated by the modifiers. This is beyond the focus and scope of this study, which is to elucidate the fundamental translation properties of the repeat RNA at the single molecule level. Nevertheless, we inspected the published data from AnswerALS (Baxi, 2022 Nat Neurosci) and compared the genetic modifiers that we identify from our mechanistic study. From the proteomic study of patient iPSC-neurons, we noticed that the protein level of Pelota in C9-ALS patients is significantly lower compared to healthy individuals (control group). Pelota plays a significant role in the ribosome-associated quality control and we showed it can modulate repeat translation elongation. We added this data to Figure 6 and edited in discussion.

Minor comments:

The arrangement of some of the figures does not correspond to the text. For example, Fig. 1f appears before Fig. 1a in the text. Such arrangements increased the difficulty of reading the manuscript.

Response: We thank the reviewer for those remarks. We have now adjusted the order of figures and the description.

Reviewer #3 (Remarks to the Author):

In this work, several insights are made regarding DPR synthesis from G4C2 repeats. Sophisticated techniques, including single molecule translation imaging, are used appropriately and rigorously to assess facets of DPR initiation and elongation. More traditional, “bulk” luciferase-based methods are used to confirm their findings. The CRISPR screen of 1906 proteins identified RQC factors involved in DPR translation and the findings supported through functional assays. Overall, the methodologies are elegant, the experimental design is rigorous, and interpretations of data are well-reasoned. This is a nice, scientifically sound mechanistic study.

With that said, DPRs have been studied extensively and the current findings are incremental. The findings add little new information and mostly confirm what has already been observed. They are thus of nominal significance to the field, in the reviewer’s opinion. For example, the observation that frame-shifting can occur had already been made, reducing novelty of the current study. The identification of ZNF598 and Pelota as modulators of DPR translation is interesting, but there is little evidence or discussion of how the mechanism could potentially be targeted as a therapeutic approach. It is frustrating that the contribution of DPRs to disease remains controversial and that the current study does not reduce ambiguity.

Response: We greatly appreciated the reviewer’s recognition of the scientific soundness of this study. We could do a better job to delineate the impact of the work. The potential toxicity of DPR has been investigated in several studies in the literature, and it is not the focus of this work. We focused on the biogenesis of DPR. The translation properties of the repeat RNA have never been directly measured. The previous literature always relied on the bulk assays measuring the final protein product, which can be influenced by many factors besides translation. Although it has been shown frame-shifting can occur, it cannot be quantified how frequently it can occur. Our work for the first time provides direct evidence to study the various translation steps of expanded repeats, one key step of the toxic DPR biogenesis. It is equally important as DPR toxicity studies, and provides insights on the basic molecular mechanism relevant to the disease from a different angle. First, we showed the CUG near-cognate start codon contributes to the initiation of the GA frame, but not GP or GR frame, when measuring translation initiation directly. Second, we proved that frameshift occurs, but infrequently during the translation of C9ORF72 repeats and the frequency among different frames is not equal. Third, we showed that the translation elongation speeds in GP and GR frames are slower than in GA frame. Fourth, the amino acid composition of DPRs affects the translation elongation speed, thereafter influences the dipeptide production in patient cells. Fifth, RQC factors modulate the translation dynamics of DPRs.

To provide potential therapeutic targets through our mechanistic study, we looked further into the published C9ALS/FTD patient sample. We found that C9 ALS patients have lower Pelota protein levels compared to healthy individuals. The CRISPR screen identified a series of other genetic modifiers that modulate the production of the most toxic poly-GR, which is worthwhile to follow up in the future. But it

is out-of-scope for the current study. This study clarified ambiguity of how translation occurs on a repeat sequence by directly visualizing the translation on single RNA molecules, which cannot be revealed by ensemble approaches. It is not only important for the GGGGCC repeat, but also shed light on other repeat expansion diseases. We believe the novelty and significance of the findings on the molecular mechanism of repeat RNA translation is high and also provides important information when understanding diseases.

REVIEWERS' COMMENTS

Reviewer #1 (Remarks to the Author):

The authors nicely answered my queries and I am satisfied with their responses. I think this is a very interesting study and worthy of publication.

Reviewer #2 (Remarks to the Author):

The authors have significantly improved the quality of the manuscript. The following discussion points should be included in the discussion:

Licata et al. (2022) EMBO J. 41: e105026 showed that protein kinase A modulates DPR protein levels in G4C2 models, including C9ALS/FTD patient-derived iPSC motor neurons. The authors should discuss how PKA may regulate DPR protein translation/elongation. Did the authors identify any PKA pathway component from their genome-wide CRISPR-Cas9 knockout screen?

When discussing the relative translation level of the DPR proteins, the authors should make reference to DPR protein levels determined in patient-derived iPSC motor neurons (e.g. Licata et al.). Does the expression of GA being the highest in neurons?

Discuss how their findings may be relevant to RAN translation control in other repeat expansion diseases.

Reviewer #3 (Remarks to the Author):

In the updated manuscript, substantive changes have been made in response to the reviewer's concerns. The authors have now addressed most of the technical and aesthetic concerns. Generally, the scientific findings are sound and align with previous reports. The use of sophisticated techniques is commendable and the identification of ZNF598 and Pelota as modulators of DPR translation is novel. Despite these attributes, however, the narrow scope and limited significance of this work are problematic, in the reviewer's opinion. While the current study provides more detailed knowledge of the role of the CUG codon, frame-shift frequencies, and elongation dynamics, these findings mostly replicate established biology. [e.g. Green, K. M. et al. Nature communications 8, 1-13 (2017); Tabet, R. et al. Nature communications 9, 1-14 (2018)]. Further, some important technical limitations obfuscate disease relevance. For example, reporter constructs with only 70 repeats are used in this study whereas the endogenous mutation is much longer and can vary in size. Second, while multiple cell lines are used to increase rigor, the findings are not validated in relevant cell types (i.e. patient-derived neurons & glia) or primary tissues. Therefore, while this is a sophisticated mechanistic study, its potential impact on the field is deemed to be moderately low.

Response to reviewers

We would like to thank all the reviewers for recognizing our efforts in improving the paper. We now responded to the remaining comments point by point. We also made minor changes to the paper to address the reviewer's comments.

Reviewer #2 (Remarks to the Author):

The authors have significantly improved the quality of the manuscript. The following discussion points should be included in the discussion:

Licata et al. (2022) EMBO J. 41: e105026 showed that protein kinase A modulates DPR protein levels in G4C2 models, including C9ALS/FTD patient-derived iPSC motor neurons. The authors should discuss how PKA may regulate DPR protein translation/elongation. Did the authors identify any PKA pathway component from their genome-wide CRISPR-Cas9 knockout screen?

Response: We thank the reviewer for recognizing the improved quality. In the CRISPR screen, we did not target the whole genome. Our library only contains gRNAs targeting 1906 RNA binding proteins and ribosome proteins. We described the library in the Result, but also had a mistake in writing "genome-wide" screen. We apologize for the confusion and have now corrected it in the manuscript.

When discussing the relative translation level of the DPR proteins, the authors should make reference to DPR protein levels determined in patient-derived iPSC motor neurons (e.g. Licata et al.). Does the expression of GA being the highest in neurons? Discuss how their findings may be relevant to RAN translation control in other repeat expansion diseases.

Response: We thank the reviewer for the suggestion. Now, we discussed these important points in the discussion (highlighted). We have referenced the publication from Licata et al. Poly-GA reporter expression is indeed the highest in neurons, as shown previously [Green K.M. et al, Nat Comm 2017 ,PMID 29222490; Westergard T. et al, EMBO Mol Med, 2019, PMID 30617154]. We have cited these references in the manuscript.

Reviewer #3 (Remarks to the Author):

In the updated manuscript, substantive changes have been made in response to the reviewer's concerns. The authors have now addressed most of the technical and aesthetic concerns. Generally, the scientific findings are sound and align with previous reports. The use of sophisticated techniques is commendable and the identification of ZNF598 and Pelota as modulators of DPR translation is novel. Despite these attributes, however, the narrow scope and limited significance of this work are problematic, in the reviewer's opinion. While the current study provides more detailed knowledge of the role of the CUG codon, frame-shift frequencies, and elongation dynamics, these findings mostly replicate established biology. [e.g. Green, K. M. et al. Nature communications 8, 1-13 (2017); Tabet, R. et al. Nature communications 9, 1-14 (2018)]. Further, some important technical limitations obfuscate disease relevance. For example, reporter constructs with only 70 repeats are used in this study whereas the endogenous mutation is much longer and can vary in size. Second, while multiple cell lines are used to increase rigor, the findings are not validated in relevant cell types (i.e. patient-derived neurons & glia) or primary tissues. Therefore, while this is a sophisticated mechanistic study, its potential impact on the field is deemed to be moderately low.

Response: We thank the reviewer's recognition of our efforts to substantively improve the manuscript to address his/her concerns and acknowledgment of our "sophisticated mechanistic study". We have pointed out our significant contribution to the field in the previous response to the reviewer. Particularly, our findings have considerable differences from the references the reviewer cited (Green et al 2017 and Tabet et al 2018). First, although we agreed with the two papers about the role of the -24nt CUG pseudo start codon for initiation in the GA frame, we found that the initiation alone is not sufficient to explain the observed difference in different frames. Second, we showed that there is a higher propensity for frameshifting from GR to GA than the reverse. It is unlikely that GR is predominantly generated by frameshifting from GA initiated at the CUG frame. Third, we demonstrated that the elongation speeds between different frames are drastically different. The slow elongation in the GR frame may lead to ribosome collision and ribosome-associated quality control. Fourth, the reviewer pointed out that our reporters have only 70 repeats, and may be shorter than the one in the patient. The shorter repeat is purely due to the difficulty in producing long repeats experimentally. Similar or shorter repeats are broadly used in previous publications, including the one cited by the reviewer. This showed that we could derive interesting and important insights from these moderate-length repeats. Finally, although we have yet to perform these experiments in patient tissue because of the technical difficulty of live cell single molecule imaging, we believe that the fundamental molecular mechanism is universal and our contribution is impactful to the field.